# Chiral Gauge Theories on $R^3 \times S^1$ and SUSY Breaking

**Jun Seok Lee and John Terning**

*Center for Quantum Mathematics and Physics (QMAP)*
*Department of Physics, University of California*
*Davis CA 95616*

phylee@ucdavis.edu, jterning@gmail.com

## Abstract

We study $SU(5)$ chiral gauge theories on $R^3 \times S^1$. With an unequal number of fundamental and antifundmental matter representations we calculate nontrivial pre-ADS superpotentials generated by composite multi-monopoles. We also point out that the structure of the composite multi-monopoles can be determined simply from the affine Dynkin diagrams of the gauge group and its unbroken subgroup. For the case of one flavor, we find that the superpotential is independent of the composite meson. We show that dynamical 4D SUSY breaking in the simplest chiral $SU(5)$ gauge theory can be demonstrated directly via semi-classical effects on the circle.

## 1 Introduction

Compactifying on a circle (4D $\rightarrow$ $R^3 \times S^1$) provides an intriguing approach to understanding strongly coupled supersymmetric (SUSY) gauge theories: holomorphic quantities can be calculated at weak coupling on a sufficiently small circle and continued back to the 4D limit [1–8]. In the compactified theory an adjoint vacuum expectation value (VEV) typically breaks the non-Abelian gauge group to $U(1)$ factors, and the low-energy dynamics boils down to understanding the resulting monopole/instanton solutions [9] and their zero modes. A monopole with exactly two fermionic zero modes generates a semi-classical term in the superpotential since it corresponds to a dynamical mass insertion amplitude. In the absence of matter fields this describes the low-energy dynamics of a Coulomb branch moduli space [10]. Adding matter fields allows for richer behavior on the mixed Higgs-Coulomb branch. On the mixed branch [11] matter VEVs can break some of the $U(1)$'s down to diagonal $U(1)$ subgroups, thus confining some monopoles [12] via Nielsen-Olesen flux tubes [13, 14]. Confined multi-monopole configurations can also contribute to the (pre-ADS) superpotential [11], and these multi-monopole configurations have the correct topological charges to be the monopoles of the unbroken non-Abelian gauge group that is recovered in the 4D limit. For $SU(N)$ with $F$ flavors, the Affleck-Dine-Seiberg (ADS) superpotential [15, 16] was discovered long ago, but there is no reliable dynamical calculation for $F < N-1$ in 4D. On the circle the matter VEVs reduce the rank of the gauge group, producing confined multi-monopoles and a corresponding (pre-ADS) superpotential [11]. Integrating out massive modes and taking the 4D limit gives exactly the ADS superpotential [11].

The inclusion of antisymmetric matter allows for more complicated breaking patterns, e.g. $SU(2N)$ can break to $Sp(2N)$, and in some cases breaking the gauge group does not reduce the rank. If the rank is not reduced the corresponding monopoles are not confined but may still form

bound states [17–19]. Antisymmetric matter also allows for the construction of a chiral gauge theory, with $SU(5)$ providing the standard example. So far the only chiral gauge theory studied on $R^3 \times S^1$ is $SU(2)$ with a chiral superfield in the four dimensional representation [20]. It was shown that no superpotential is generated by monopoles, suggesting that the SUSY breaking conjectured [21] for this model does not occur.

Here we study chiral $SU(5)$ gauge theories on $R^3 \times S^1$. We explore the pattern of symmetry breaking, monopole confinement, and the resulting superpotential for a variety of matter content, using chiral superfields in the following representations of $SU(5)$: $\yng(1,1) = \mathbf{10}$, $\overline{\yng(1,1)} = \overline{\mathbf{10}}$, $\square = \mathbf{5}$, and $\overline{\square} = \overline{\mathbf{5}}$. In section 2 we give an overview of the models and a quick summary of the results for superpotentials on $R^4$, $R^3 \times S^1$, and $R^3$. We review $SU(5)$ monopole solutions and their zero modes in section 3. We analyze a vector-like $SU(5)$ theory with antisymmetric matter as a warm-up example in section 4. In section 5 we discuss chiral $SU(5)$ theories with $\yng(1,1) + F\square + (F+1)\overline{\square}$ with $F > 0$ on $R^3 \times S^1$. We also explore SUSY breaking for $SU(5)$ with $\yng(1,1)$ and $\overline{\square}$ on the circle so that it can be analyzed at weak coupling in section 6. Finally we present our conclusions in Section 7. We also provide a review of zero mode configurations on monopoles in Appendix A, and Coulomb branches and their moduli in Appendix B. Appendix C discusses the dynamics of chiral $SU(5)$ theories in fundamental Weyl chamber regions outside the region discussed in the main text. Appendix D examines the case of $F = 2$ in alternative regions of the moduli space, demonstrating continuity of the low-energy physics as boundaries between regions are crossed.

## 2  Results

We mainly interested in $SU(5)$ gauge theories with chiral superfields in three different representations: one antisymmetric tensor, $F$ fundamentals, and $F + 1$ antifundamentals. The gauge and global charges of the fields and their gauge invariant composites are given in Table 1.

| | $SU(5)$ | $SU(F)$ | $SU(F+1)$ | $U(1)_1$ | $U(1)_2$ | $U(1)_R$ |
|---|---|---|---|---|---|---|
| $A$ | $\yng(1,1)$ | $1$ | $1$ | $0$ | $2F+1$ | $0$ |
| $Q$ | $\square$ | $\square$ | $1$ | $-F-1$ | $-3$ | $2$ |
| $\overline{Q}$ | $\overline{\square}$ | $1$ | $\square$ | $F$ | $-3$ | $-\frac{6}{F+1}$ |
| $M = Q\overline{Q}$ if $F \geq 1$ | | $\square$ | $\square$ | $-1$ | $-6$ | $2 - \frac{6}{F+1}$ |
| $B_2 = A^2 Q$ if $F \geq 1$ | | $\square$ | $1$ | $-F-1$ | $4F-1$ | $2$ |
| $\overline{B}_1 = A\overline{Q}^2$ if $F \geq 1$ | | $1$ | $\yng(1,1)$ | $2F$ | $2F-5$ | $-\frac{12}{F+1}$ |
| $B_1 = AQ^3$ if $F \geq 3$ | | $1,\overline{\square},\overline{\yng(1,1)},\ldots$ | $1$ | $-3(F+1)$ | $2F-8$ | $6$ |
| $\overline{B}_0 = \overline{Q}^5$ if $F \geq 4$ | | $1$ | $1,\overline{\square},\overline{\yng(1,1)},\ldots$ | $5F$ | $-15$ | $-\frac{30}{F+1}$ |

Table 1: Gauge and global quantum numbers of the fields. Top rows: elementary matter fields. Bottom rows: gauge invariant composite fields.

A summary of the superpotentials we find is given in Table 2. The vector like case with $\yng(1,1) + \overline{\yng(1,1)}$ matter has four types of composite monopoles, and hence four terms in the pre-ADS superpotential. In 4D this means there is a runaway branch and a branch with a vanishing superpotential. The 4D superpotential for $F = 4$ is well-known since it is the s-confining case [22]. The largest value of $F$ we

will consider in detail is $F = 3$ which has a deformed moduli space in 4D. The $F = 2$ superpotential on $\mathrm{R}^3 \times S^1$ results from a composite of all the $SU(5)$ monopoles, including the Kaluza-Klein (KK) monopole [4, 23], which corresponds to a single instanton in 4D and a deformed moduli space in 3D. The $F = 1$ superpotential on $\mathrm{R}^3 \times S^1$ results from a KK monopole and a composite of the other monopoles, which turns out to be independent of the composite meson field, $M$. The meson branch for $F = 1$ is completely lifted. For the SUSY breaking case, $F = 0$, there are contributions to the scalar potential from monopole-antimonopole pairs, including a KK monopole, a composite of four monopoles, and a composite of four monopoles bound to the KK monopole; the D-terms lift the vacuum in this case, while in 3D there is simply a Coulomb-branch-runaway superpotential.

| matter | 4D | $R^3 \times S^1$ | 3D |
|---|---|---|---|
| $\yng(1,1) + \overline{\yng(1,1)}$ | $2(\epsilon_a + \epsilon_b)\sqrt{\frac{\Lambda^{12}}{T^3}}$ | $\eta Y + \frac{1}{YT^3} + \eta\frac{Y_2 Y_3}{T^2} + \frac{1}{Y_2 Y_3 T}$ | $\frac{1}{YT^3} + \frac{1}{Y_2 Y_3 T}$ |
| $\yng(1,1) + 3\,\square + 4\,\overline{\square}$ | $X\left(B_2\overline{B}_1 M^2 - B_1\overline{B}_1^2 - \Lambda^{10}\right)$ | $Y\left(B_2\overline{B}_1 M^2 - B_1\overline{B}_1^2 - \eta\right)$ | $Y\left(B_2\overline{B}_1 M^2 - B_1\overline{B}_1^2\right)$ |
| $\yng(1,1) + 2\,\square + 3\,\overline{\square}$ | $\frac{\Lambda^{11}}{B_2\overline{B}_1 M}$ | $\frac{\eta}{B_2\overline{B}_1 M}$ | $\lambda\left(Y(B_2\overline{B}_1 M) - 1\right)$ |
| $\yng(1,1) + \square + 2\,\overline{\square}$ | $2\,\epsilon\frac{\Lambda^6}{\sqrt{B_2\overline{B}_1}}$ | $\eta Y + \frac{1}{YB_2\overline{B}_1}$ | $\frac{1}{YB_2\overline{B}_1}$ |
| $\yng(1,1) + \overline{\square}$ | SUSY breaking | SUSY breaking | Coulomb branch runaway |

Table 2: Superpotentials for various mixed Higgs-Coulomb branches of $SU(5)$ gauge theories on $R^4$, $R^3 \times S^1$, and $R^3$, where $T = A\overline{A}$ and $\epsilon = \pm 1$.

# 3   Monopoles of $SU(5)$

At a simplistic level, compactifying a non-Abelian gauge theory on a circle converts the gauge field component along the circle to a scalar adjoint which can obtain a VEV. If this VEV breaks the gauge group down to one containing $U(1)$ factors then there are necessarily monopole solutions [9].

To write out the monopole solutions one must first chose a set of Cartan generators corresponding to the $U(1)$ subgroups [24]. The standard basis of the Cartan subalgebra of $SU(5)$ is given by:

$$H_1 = \frac{1}{2}\mathrm{diag}(1, -1, 0, 0, 0) , \tag{3.1}$$

$$H_2 = \frac{1}{2\sqrt{3}}\mathrm{diag}(1, 1, -2, 0, 0) , \tag{3.2}$$

$$H_3 = \frac{1}{2\sqrt{6}}\mathrm{diag}(1, 1, 1, -3, 0) , \tag{3.3}$$

$$H_4 = \frac{1}{2\sqrt{10}}\mathrm{diag}(1, 1, 1, 1, -4) , \tag{3.4}$$

which we can assemble into a vector

$$\mathbf{H} = (H_1, H_2, H_3, H_4) . \tag{3.5}$$

It will be convenient to use the simple roots

$$\alpha_1 = (1, 0, 0, 0) , \tag{3.6}$$

$$\alpha_2 = \left(-\frac{1}{2}, \frac{\sqrt{3}}{2}, 0, 0\right) , \tag{3.7}$$

$$\alpha_3 = \left(0, -\frac{1}{\sqrt{3}}, \sqrt{\frac{2}{3}}, 0\right) , \tag{3.8}$$

$$\alpha_4 = \left(0, 0, -\frac{\sqrt{3}}{2\sqrt{2}}, \frac{\sqrt{5}}{2\sqrt{2}}\right) . \tag{3.9}$$

The corresponding Cartan generators are:

$$Q_1 = \alpha_1 \cdot \mathbf{H} = \frac{1}{2}\text{diag}(1, -1, 0, 0, 0) , \tag{3.10}$$

$$Q_2 = \alpha_2 \cdot \mathbf{H} = \frac{1}{2}\text{diag}(0, 1, -1, 0, 0) , \tag{3.11}$$

$$Q_3 = \alpha_3 \cdot \mathbf{H} = \frac{1}{2}\text{diag}(0, 0, 1, -1, 0) , \tag{3.12}$$

$$Q_4 = \alpha_4 \cdot \mathbf{H} = \frac{1}{2}\text{diag}(0, 0, 0, 1, -1) . \tag{3.13}$$

The static $SU(2)$ monopole solution can simply be embedded [25] in $SU(N)$. We first write the asymptotic value of the adjoint scalar along the $z$-axis as

$$\lim_{z \to \infty} \phi = v \, \mathbf{h} \cdot \mathbf{H} , \tag{3.14}$$

where $\mathbf{h}$ is a unit vector.

For each simple root $\alpha_i$ there is an $SU(2)$ subgroup whose diagonal generator is

$$\tau_i^3 = \alpha_i \cdot \mathbf{H} . \tag{3.15}$$

The basis of simple roots can be chosen such that

$$\mathbf{h} \cdot \alpha_i \geq 0 . \tag{3.16}$$

The region of adjoint VEVs that satisfies (3.16) for a fixed set of simple roots is called the fundamental Weyl chamber. Then we can write the monopole solution [24] associated with the $i$th root as

$$\phi = \hat{r}^a \tau_i^a v(\mathbf{h} \cdot \alpha_\mathbf{i}) f(r, v(\mathbf{h} \cdot \alpha_\mathbf{i})) + v(\mathbf{h} - (\mathbf{h} \cdot \alpha_\mathbf{i})\alpha_\mathbf{i}) \cdot \mathbf{H} , \tag{3.17}$$

where, $\tau_i^a$ are generators ($a = 1, 2, 3$) of the $SU(2)$ subgroup associated with $\alpha_i$.

In a given Weyl chamber we can decompose the adjoint VEV into a piece that acts like an adjoint of the $SU(2)$ subgroup, given by

$$v(\mathbf{h} \cdot \alpha_\mathbf{i})(\alpha_\mathbf{i} \cdot \mathbf{H}) \tag{3.18}$$

and a remainder that acts like a singlet under the $SU(2)$ subgroup, given by

$$v(\mathbf{h} - (\mathbf{h} \cdot \alpha_\mathbf{i})\alpha_\mathbf{i}) \cdot \mathbf{H} . \tag{3.19}$$

The magnetic field associated with the monopole (3.17) is [24]

$$\mathbf{B} = \frac{g_i\,\hat{r}}{er^2} \tag{3.20}$$

where $e$ is the electric coupling constant and the magnetic charge is given in terms of the dual root vector $\alpha_i^*$:

$$g_i = \alpha_i^* \cdot \mathbf{H} \ , \qquad \alpha_i^* = \frac{\alpha_i}{\alpha_i^2} \tag{3.21}$$

For the case of $SU(N)$, the dual root vector simplifies to $\alpha_i^* = \alpha_i$ .

Compactifying onto $R^3 \times S^1$, the component of the gauge field along the $S^1$ direction plays the role of the adjoint scalar, and all the static monopole solutions continue to be solutions with the spatial dependence entirely in $R^3$. There is a fifth monopole as well, which is constructed by performing a periodic gauge transformation along the $S^1$ that takes the adjoint back to itself after one period. This is the twisted, or KK, monopole [23, 26]. It is associated with the lowest root

$$\alpha_0 = -\alpha_1 - \alpha_2 - \alpha_3 - \alpha_4 \ . \tag{3.22}$$

The simple roots along with $\alpha_0$ can be used to construct the affine (extended) Dynkin diagram, which will be useful to us in what follows.

# 4 Waru-up Example: $SU(5)$ with $\square\!\square + \overline{\square\!\square}$

Let us first consider a non-chiral theory: $SU(5)$ with chiral supermultiplets $A$ and $\overline{A}$, which transform as the antisymmetric, $\square\!\square$, and its conjugate, $\overline{\square\!\square}$, under $SU(5)$. At a generic point on the classical moduli space the antisymmetric VEVs can be gauge rotated to

$$A = \begin{pmatrix} 0 & 0 & 0 & 0 & a \\ 0 & 0 & 0 & b & 0 \\ 0 & 0 & 0 & 0 & 0 \\ 0 & -b & 0 & 0 & 0 \\ -a & 0 & 0 & 0 & 0 \end{pmatrix} \ , \qquad \overline{A} = \begin{pmatrix} 0 & 0 & 0 & 0 & a \\ 0 & 0 & 0 & b & 0 \\ 0 & 0 & 0 & 0 & 0 \\ 0 & -b & 0 & 0 & 0 \\ -a & 0 & 0 & 0 & 0 \end{pmatrix} \ . \tag{4.1}$$

$D$-flatness requires $A^\dagger A - \overline{A}^\dagger\overline{A} = 0$. With this matter content the 4D one-loop $\beta$ function coefficient is

$$b_1 = 3\,T(\mathbf{Ad}) - T\left(\square\!\square\right) - T\left(\overline{\square\!\square}\right) = 3 \cdot 5 - 2 \cdot \frac{5-2}{2} = 12 \ , \tag{4.2}$$

where $T(\mathbf{R})$ is the index of the representation $\mathbf{R}$.

With $a = 0$, the antisymmetric VEV $b$ breaks $SU(5) \to SU(3)_a \times SU(2)_b$ with Cartan generators

$$Q_{1+2} = \frac{1}{2}\,\mathrm{diag}(1,0,-1,0,0) \ , \tag{4.3}$$

$$Q_{3+4} = \frac{1}{2}\,\mathrm{diag}(0,0,1,0,-1) \ , \tag{4.4}$$

$$Q_{2+3} = \frac{1}{2}\,\mathrm{diag}(0,1,0,-1,0) \ , \tag{4.5}$$

while the broken $U(1)$ generator is

$$X = 2(Q_1 - Q_4) - Q_2 + Q_3 = \frac{1}{2} \operatorname{diag}(2, -3, 2, -3, 2) . \tag{4.6}$$

$Q_{1+2}$ and $Q_{3+4}$ are the Cartan elements of the $SU(3)_a$ while $Q_{2+3}$ is the Cartan element of $SU(2)_b$. The scales of the $SU(5)$ theory and the low-energy $SU(3)_a \times SU(2)_b$ theory are related by

$$\Lambda_{(3a)}^8 = \frac{\Lambda^{12}}{b^4} , \qquad \Lambda_{(2b)}^6 = \frac{\Lambda^{12}}{b^6} . \tag{4.7}$$

The antisymmetric decomposes as $A \sim 10 \to (3,2) + (\bar{3},1) + (1,1)$. The $(3,2)$ and $(\bar{3},2)$ are eaten by the broken gauge bosons, so the low-energy theory has one flavor for $SU(3)_a$ and no flavors for $SU(2)_b$. The VEV $b \neq 0$ classically lifts part of the Coulomb moduli. In other words, on this mixed Higgs-Coulomb branch there are additional restrictions on the $SU(5)$ adjoint VEV,

$$\phi = \operatorname{diag}(v_1, v_2, v_3, v_4, v_5) . \tag{4.8}$$

Working in one particular region of the fundamental Weyl chamber (see Table 8 in Appendix B):

$$|v_5| \geq v_1 \geq |v_4| \geq v_2 \geq v_3 \geq 0 \geq v_4 \geq v_5 , \tag{4.9}$$

we must further satisfy

$$v_1 + v_3 + v_5 = 0 , \qquad v_2 + v_4 = 0 , \tag{4.10}$$

that is, the VEVs take the form of the adjoint VEVs of the low-energy gauge group $SU(3)_a \times SU(2)_b$, and can be expanded in the basis $Q_{1+2}$, $Q_{3+4}$, and $Q_{2+3}$. Eqs. (4.9) and (4.10) imply

$$|v_5| = v_1 + v_3 \geq v_2 . \tag{4.11}$$

Turning on both antisymmetric VEVs ($a$ and $b$) breaks $SU(5) \to SU(2)_a \times SU(2)_b$ with Cartan generators

$$Q_a = Q_{1+2+3+4} = \frac{1}{2} \operatorname{diag}(1, 0, 0, 0, -1) , \tag{4.12}$$

$$Q_b = Q_{2+3} = \frac{1}{2} \operatorname{diag}(0, 1, 0, -1, 0) , \tag{4.13}$$

and there is a second broken $U(1)$ generator

$$X' = Q_1 + Q_2 - Q_3 - Q_4 = \frac{1}{2} \operatorname{diag}(1, 0, -2, 0, 1) . \tag{4.14}$$

The scales of the $SU(5)$ theory and the low-energy $SU(2)_a \times SU(2)_b$ theory are related by

$$\Lambda_{(2a)}^6 = \frac{\Lambda^{12}}{a^2 b^4} , \qquad \Lambda_{(2b)}^6 = \frac{\Lambda^{12}}{b^6} . \tag{4.15}$$

There are now further restrictions on the $SU(5)$ adjoint VEV $\phi$. At a generic point on the mixed Higgs-Coulomb branch parametrized by (4.1) the adjoint VEV is restricted to have the form

$$\phi = \operatorname{diag}(v_1, v_2, 0, -v_2, -v_1) , \tag{4.16}$$

so we have VEVs corresponding to the adjoints of the unbroken gauge group $SU(2)_a \times SU(2)_b$. In other words, we are forced to be on the boundary of the region (4.9). We can approach this boundary, for example, by taking

$$\phi = \text{diag}(v_1, v_2, 2\epsilon, -v_2 - \epsilon, -v_1 - \epsilon) \;, \tag{4.17}$$

satisfying the fundamental Weyl chamber conditions

$$v_1 > v_2, \; v_2 > v_3 = 2\epsilon, \quad v_3 > v_4 \Rightarrow v_2 > -3\,\epsilon, \quad v_4 > v_5 \Rightarrow v_1 > v_2 \;, \tag{4.18}$$

and finally taking the limit $v_3 = 2\epsilon \to 0^+$. In this region of the moduli space, the zero mode condition (see Eq. A.11 in the Appendix) for the $k$th doublet $(A_{i,k}, A_{i+1,k})$ from the antisymmetric tensor on the $i$th BPS monopole shows that $(A_{1,4}, A_{2,4})$, $(A_{3,2}, A_{4,2})$ and $(A_{4,1}, A_{5,1})$ have fermionic zero modes on monopoles 1, 3, and 4 respectively. The conjugate representation, $\overline{A}$, has the same distribution of zero modes.

The $U(1)$ charges of the 4 BPS monopoles and the KK monopole are given in Table 3. The charge of the monopole under $Q_X$, for example, can be calculated from (3.21) via $\text{Tr}\, g_i Q_X$. The

| monopole | $Q_1$ | $Q_2$ | $Q_3$ | $Q_4$ | $Q_a$ | $Q_b$ | $Q_X$ | $Q_{X'}$ |
|:--------:|:-----:|:-----:|:-----:|:-----:|:-----:|:-----:|:-----:|:--------:|
| 1 | 1 | 0 | 0 | 0 | 1 | -1 | $\frac{5}{2}$ | $\frac{1}{2}$ |
| 2 | 0 | 1 | 0 | 0 | 1 | 1 | $-\frac{5}{2}$ | 1 |
| 3 | 0 | 0 | 1 | 0 | 1 | 1 | $\frac{5}{2}$ | -1 |
| 4 | 0 | 0 | 0 | 1 | 1 | -1 | $-\frac{5}{2}$ | $-\frac{1}{2}$ |
| KK | -1 | -1 | -1 | -1 | -4 | 0 | 0 | 0 |

Table 3: Charges of the various $SU(5)$ monopoles on $R^3 \times S^1$.

structure of the low-energy effective theories and the resulting composite monopoles is nicely summarized in the affine (extended) Dynkin diagrams for $SU(5)$ and its subgroups as shown in Fig. 1.

Let's take a look at the low-energy effective theory and the structure of the composite monopoles in detail. Turning on only the $b$ VEV produces various types of composite monopoles that are neutral under the broken $U(1)_X$ (since $X$ charges are confined). We are primarily interested in composites that have two unlifted fermion zero modes, since these are the only monopoles that contribute to the low-energy effective superpotential [3]. There are four types of confined composite monopoles: monopole 1 with monopole 2, monopole 2 with monopole 3, monopole 3 with monopole 4, and monopole 1 with monopole 4. The KK monopole itself is also neutral under the broken $U(1)_X$. However, as the extended Dynkin diagram for $SU(2)_b$ in Fig. 1 shows, the KK monopole and the confined 1+4 composite monopole must combine together in order to serve as an effective KK monopole for the $SU(2)_b$ of the low-energy effective theory. This indicates that there is an attractive force that is sufficiently strong for the KK monopole to form a bound state [17,18] with monopoles 1 and 4. Notice that the KK monopole and the 1+4 composite have the opposite charge under the unbroken generator $Q_a$. The KK monopole must also appear by itself for the $SU(2)_a$ low-energy gauge group, as shown in Fig. 1. In the presence of the $b$ VEV the background adjoint

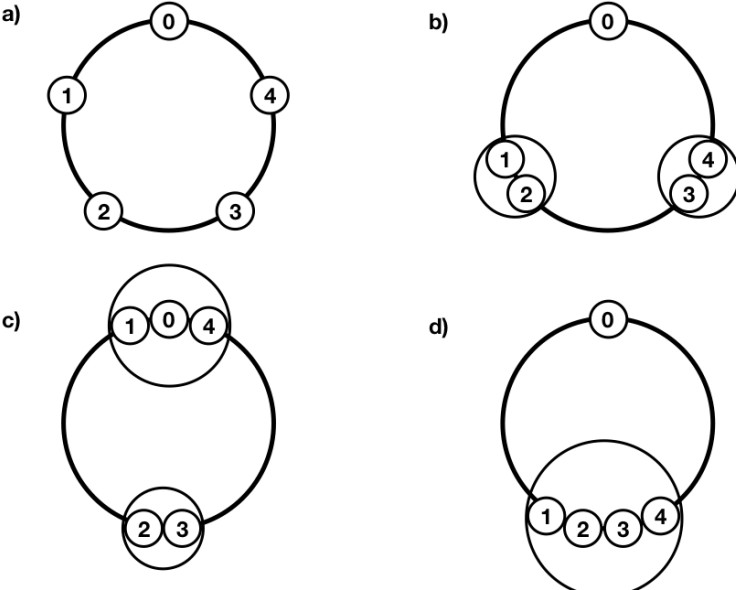

Figure 1: The extended Dynkin diagrams for $SU(5)$ and its subgroups: a) $SU(5)$, b) $SU(3)_a$, c) $SU(2)_b$, and d) $SU(2)_a$. Each node represents a simple root, or the lowest root (indicated by 0). Single lines between roots indicated roots separated by 120°, the absence of lines connecting roots means that they are orthogonal. The double lines for $SU(2)$ indicate that the ordinary simple root and the lowest negative root are anti-parallel, since, with one Cartan element the weight space is only one-dimensional.

VEV (4.8) can be split into two pieces:

$$\phi = \text{diag}(v_1, 0, v_3, 0, -v_1 - v_3) + \text{diag}(0, v_2, 0, -v_2, 0) \tag{4.19}$$
$$\equiv \widetilde{\phi}_{3a} + \widetilde{\phi}_{2b} , \tag{4.20}$$

where $\widetilde{\phi}_{3a}$ corresponds to an adjoint under the $SU(3)_a$ and a singlet under the $SU(2)_b$, and $\widetilde{\phi}_{2b}$ corresponds to a singlet under the $SU(3)_a$ and an adjoint under the $SU(2)_b$. Under $SU(3)_a$, the condition $v_3 > 0 > -v_1 - v_3$ shows that all matter zero modes live on the second monopole of $SU(3)_a$, which is the composite of monopole 3 with monopole 4. The 3+4 composite thus has four fermion zero modes (there are two $SU(3)_a$ gaugino zero modes as well) and does not contribute to the superpotential. That this is consistent can be seen by noting that two matter zero modes on monopole 1 are lifted together with two gaugino zero modes by the Yukawa coupling,

$$g \, A^{*\,4,2} \lambda_2^2 \, \psi_{2,4} + h.c. , \tag{4.21}$$

where $\psi_{i,j}$ represents the fermion component of the antisymmetric matter field $A$. Thus only two gaugino zero modes are left in the 1+2 composite monopole. Similarly, two matter zero modes on monopole 3 are lifted along with two gaugino zero modes by the Yukawa coupling,

$$g \, A^{*\,2,4} \lambda_4^4 \, \psi_{4,2} + h.c. , \tag{4.22}$$

however, two matter zero modes on the monopole 4 remains unlifted, which makes a tally of four zero modes on the 3+4 composite. One can similarly check that the 2+3 composite and the

KK+1+4 bound state have only two gaugino zero modes, so they contribute to the superpotential. A sketch of each multi-monopole composite under $SU(3)_a \times SU(2)_b$ is shown in Fig. 2. For multi-monopole composite diagrams throughout the paper we note that the fermion zero mode can propagate along the flux-tube/string when monopoles are confined [27] and we indeed move the zero modes to simplify the "resonance" diagrams.[1] See Appendix C for more details.

The effective superpotential is[2]:

$$W = \eta Y_1 Y_2 Y_3 Y_4 + \frac{1}{Y_1 Y_2 A\overline{A}} + \eta \frac{Y_2 Y_3}{A^2 \overline{A}^2} + \frac{1}{Y_2 Y_3 A\overline{A}} \ . \tag{4.23}$$

It is conventional to drop the dependence on the radius $R$ in the coefficients. Note that the first term in (4.23) is just the single KK monopole, while the third term arises from the composite of the KK monopole with monopole 1 and monopole 4, so that the $Y_1 Y_4$ dependence cancels between numerator and denominator.

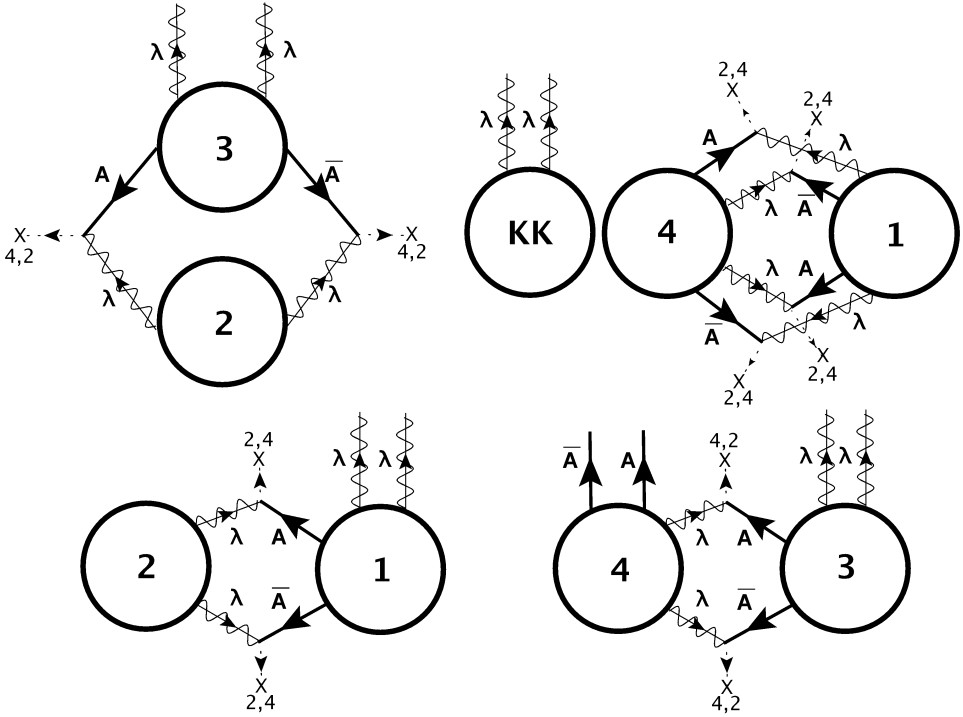

Figure 2: A sketch of the multi-monopole composites in the low-energy $SU(3)_a \times SU(2)_b$ theory. The antisymmetric VEVs are represented by a cross with their color indices. See text for details.

This superpotential matches the low-energy effective superpotential for $SU(3)_a \times SU(2)_b$ with matter in $(3, 1) + (\overline{3}, 1)$, which is given by

$$W = \eta_{(3a)} Y_{\mathrm{KK}}^{(3a)} + \frac{1}{Y_1^{(3a)}} + \eta_{(2b)} Y_{\mathrm{KK}}^{(2b)} + \frac{1}{Y^{(2b)}} \tag{4.24}$$

---

[1]For example, it should be understood in the 1+2 composite diagram of Fig. 2 that $\lambda_2^2$ gaugino components of monopole 1 and monopole 2 are lifted by the Yukawa coupling (4.21) and components of the gaugino zero modes associated with the unbroken generator $Q_{1+2}$ are not lifted.

[2]See Appendix B for details of Coulomb operators.

where the matching is

$$\eta^{(3a)} \;=\; \frac{\eta}{b^4}\,, \qquad \eta_{(2b)} = \frac{\eta}{b^6}\,, \qquad Y_1^{(3a)} = Y_1 Y_2\, b^2\,, \qquad Y_2^{(3a)} = Y_3 Y_4\, b^2\,, \qquad (4.25)$$

$$Y_{\mathrm{KK}}^{(3a)} \;=\; Y_1 Y_2 Y_3 Y_4\, b^4 = Y_1^{(3a)} Y_2^{(3a)}\,, \qquad Y_{\mathrm{KK}}^{(2b)} = \frac{Y_1 Y_2 Y_3 Y_4}{Y_1 Y_4}\, b^2 = Y^{(2b)} = Y_2 Y_3\, b^2\,. \qquad (4.26)$$

Note that the effective $SU(3)_a$ theory has the adjoint VEV, $\mathrm{diag}(v_1, v_3, -v_1 - v_3)$, and matter zero modes on the second monopole.

Turning on the VEV $a$ further requires that composites be neutral under $U(1)_{X'}$, since then $X'$ charges are confined. The 1+2 composite and the 3+4 composite are not neutral under $U(1)_{X'}$ but have opposite charges and thus can be confined together. The composite comprised of monopoles 1, 2, 3, and 4 has two unlifted fermion zero modes, as seen in Fig. 3, and so contributes to the low-energy superpotential.

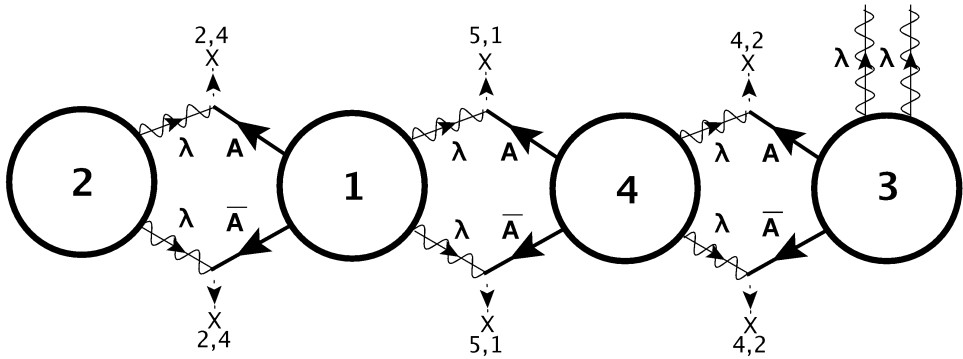

Figure 3: A sketch of the four monopole composite contributing to the superpotential in the low-energy $SU(2)_a \times SU(2)_b$. The antisymmetric VEVs are represented by a cross with their color indices.

Taking $v_3 \to 0^-$ from (4.17), we can get to the boundary of the fundamental Weyl chamber region, (4.16), from another fundamental Weyl chamber region

$$|v_5| > v_1 > |v_4| > v_2 > 0 > v_3 > v_4 > v_5\,, \qquad (4.27)$$

where there are matter zero modes on monopoles 1, 2, and 4. The difference between the two regions is that certain fermion zero modes jump from monopole 2 to monopole 3. One can see that once monopoles 2 and 3 are confined in the low-energy effective theory, the low-energy physics is smooth as we cross the boundary and we arrive at the same set of composite monopoles as described above. In fact when monopoles 2 and 3 are confined there is a Nielsen-Olesen flux-tube between them, and the fermion zero mode can propagate along the flux-tube/string [27].

Thus at a generic point on the moduli space parametrized by (4.1), we find the superpotential:

$$W \;=\; \eta Y_1 Y_2 Y_3 Y_4 + \frac{1}{Y_1 Y_2 Y_3 Y_4 (A\overline{A})^3} + \eta \frac{Y_2 Y_3}{A^2 \overline{A}^2} + \frac{1}{Y_2 Y_3 A\overline{A}}\,, \qquad (4.28)$$

which matches the low-energy effective superpotential for $SU(2)_a \times SU(2)_b$ with no matter:

$$W = \eta_{(2a)} Y^{(2a)} + \frac{1}{Y^{(2a)}} + \eta_{(2b)} Y^{(2b)} + \frac{1}{Y^{(2b)}} \qquad (4.29)$$

where the matching is given by

$$\eta_{(2a)} \;=\; \frac{\eta}{a^2\,b^4}\;, \qquad \eta_{(2b)} = \frac{\eta}{b^6}\;, \tag{4.30}$$

$$Y^{(2a)} \;=\; Y_1 Y_2 Y_3 Y_4\, a^2 b^4 = Y_1 Y_2 Y_3 Y_4\,(A\overline{A})^3\;, \qquad Y^{(2b)}\frac{Y_1 Y_2 Y_3 Y_4}{Y_1 Y_4}\,b^2 = Y_2 Y_3\, A\overline{A}\;. \tag{4.31}$$

Taking the 3D limit, $R \to 0$, we have the 3D superpotential:

$$W_{3D} = \frac{1}{Y_1 Y_2 Y_3 Y_4 (A\overline{A})^3} + \frac{1}{Y_2 Y_3 A\overline{A}} \tag{4.32}$$

which gives a runaway vacuum. Integrating out the lifted $Y_i$'s from (4.28) we can take the $R \to \infty$ limit and get the 4D superpotential:

$$W_{4D} = 2(\epsilon_a + \epsilon_b)\sqrt{\frac{\Lambda^{12}}{(A\overline{A})^3}}\;, \tag{4.33}$$

where $\epsilon_{a,b} = \pm 1$, so there are two branches of the moduli space: one with a runaway ADS superpotential, and an unlifted quantum moduli space for the meson $A\overline{A}$ with $W_{4D} = 0$.

# 5 Chiral $SU(5)$ on $R^3 \times S^1$

With the matter content $\boxed{\phantom{x}}\!\!\boxed{\phantom{x}} + F\,\square + (F+1)\overline{\square}$ for a 4D $SU(5)$ theory, the one-loop $\beta$ function coefficient is

$$b_1 = 3\,T(\mathbf{Ad}) - T\left(\boxed{\phantom{x}}\!\!\boxed{\phantom{x}}\right) - F\,T\,(\square) - (F+1)T\,(\overline{\square}) = 3\cdot 5 - \frac{5-2}{2} - \frac{F}{2} - \frac{F+1}{2} = 13 - F\;. \tag{5.1}$$

In the fundamental Weyl chamber region (4.9), there are zero modes from $\boxed{\phantom{x}}\!\!\boxed{\phantom{x}}$ on monopoles 1, 3, and 4, and one zero mode from each $\square$ and $\overline{\square}$ on monopole 3. In other words, the first fundamental monopole has two gaugino zero modes and a zero mode from a piece of the antisymmetric, namely $(A_{1,4}, A_{2,4})$ which is a doublet under the $SU(2)$ subgroup corresponding to $\alpha_1$; the second monopole has two gaugino zero modes; the third monopole has two gaugino zero modes, $F$ fundamental zero modes, $F+1$ antifundamental zero modes and a zero mode from a piece of the antisymmetric, $(A_{3,2}, A_{4,2})$, which is a doublet under the $SU(2)$ subgroup corresponding to $\alpha_3$; and the fourth monopole has two gaugino zero modes and a zero mode from a piece of the antisymmetric, $(A_{4,1}, A_{5,1})$, which is a doublet under the $SU(2)$ subgroup corresponding to $\alpha_4$. We will work with this fundamental Weyl chamber region (4.9) throughout the rest of the paper.

## 5.1 $SU(5)$ with $F = 3$: $\boxed{\phantom{x}}\!\!\boxed{\phantom{x}} + 3\square + 4\overline{\square}$

Let us first discuss the relation of this theory on $R^3 \times S^1$ to the 4D theory with $\boxed{\phantom{x}}\!\!\boxed{\phantom{x}} + (F=4)\square + 5\overline{\square}$. The 4D $F = 4$ theory is s-confining [22] and has a superpotential which can be written in terms of the gauge invariant composites (see Table 1):

$$W_{4D, F=4} \;=\; \frac{1}{\Lambda^9}\left[ B_2 M^3 \overline{B}_1 - B_1 M \overline{B}_1^2 - \overline{B}_0 B_2 B_1 \right]\;. \tag{5.2}$$

One can simply add a holomorphic (superpotential) mass term for one flavor and then integrate out that flavor to obtain the superpotential with one less flavor. In practice this means adding a

linear term for a diagonal element of the meson, $M_{44}$, and solving the equation of motion. In 4D we end up with a superpotential

$$W_{4D,F=3} = X \left( B_2 \overline{B}_1 M^2 - B_1 \overline{B}_1^2 - \Lambda^{10} \right) . \tag{5.3}$$

The equation of motion for the Lagrange multiplier $X$ produces a quantum constraint. Integrating our flavors sequentially yields the first column of Table 2.

In the compactified theory on $R^3 \times S^1$ we can add a real mass for one of the four flavors and then integrate out the massive flavor which yields the superpotential

$$W_{F=3} = Y \left( B_2 \overline{B}_1 M^2 - B_1 \overline{B}_1^2 - \eta \right) , \tag{5.4}$$

where we have identified the surviving component of the meson containing only the massive flavor with $Y$, i.e. $Y = M_{44}$. Taking the real mass $\gg 1/R$ to reach the 3D limit we have

$$W_{3D,F=3} = Y \left( B_2 \overline{B}_1 M^2 - B_1 \overline{B}_1^2 \right) . \tag{5.5}$$

## 5.2  $SU(5)$ with $F = 2$: $\Box\!\Box + 2\Box + 3\overline{\Box}$

At a generic point on the moduli space the antisymmetric VEV can be gauge rotated to

$$A = \begin{pmatrix} 0 & 0 & 0 & 0 & a \\ 0 & 0 & 0 & b & 0 \\ 0 & 0 & 0 & 0 & 0 \\ 0 & -b & 0 & 0 & 0 \\ -a & 0 & 0 & 0 & 0 \end{pmatrix} \tag{5.6}$$

which is invariant under $SU(2)_a \times SU(2)_b$, with Cartan generators $Q_a$, $Q_b$, given in Eqs. (4.12) and (4.13). $D$-flatness requires that the matter VEVs

$$Q_{f\alpha} = \begin{pmatrix} q_{1,1} & 0 \\ 0 & 0 \\ 0 & q_{2,3} \\ 0 & 0 \\ 0 & 0 \end{pmatrix} , \quad \overline{Q}_{f,\alpha}^* = \begin{pmatrix} \overline{q}_{1,1}^* & 0 & 0 \\ 0 & \overline{q}_{2,2}^* & 0 \\ 0 & 0 & 0 \\ 0 & 0 & \overline{q}_{3,4}^* \\ 0 & 0 & 0 \end{pmatrix} , \tag{5.7}$$

satisfy

$$2|a|^2 = 2|a|^2 + |q_{1,1}|^2 - |\overline{q}_{1,1}|^2 = |q_{2,3}|^2 = 2|b|^2 - |\overline{q}_{2,2}|^2 = 2|b|^2 - |\overline{q}_{3,4}|^2. \tag{5.8}$$

The matter VEVs break the entire gauge group. The gauge invariant operators for $F = 2$ are

|  | $SU(2)$ | $SU(3)$ | $U(1)_1$ | $U(1)_2$ | $U(1)_R$ |
|---|---|---|---|---|---|
| $M = Q\overline{Q}$ | $\Box$ | $\Box$ | $-1$ | $-6$ | $0$ |
| $B_2 = A^2 Q$ | $\Box$ | $1$ | $-3$ | $7$ | $2$ |
| $\overline{B}_1 = A\overline{Q}^2$ | $1$ | $\Box\!\Box$ | $4$ | $-1$ | $-4$ |

(5.9)

In 4D we have an instanton superpotential:

$$W = \frac{\Lambda^{11}}{B_2 \overline{B}_1 M} , \tag{5.10}$$

which has a runaway vacuum, so far from the origin of the moduli space SUSY is approximately restored.

**5.2.1** $F = 2$, $B_2 \gg \overline{B}_1 \gg M$

We will first consider the case with hierarchical VEVs:

$$B_2 \gg \overline{B}_1 \gg M . \tag{5.11}$$

For large matter VEVs, $A, Q, \overline{Q} \gg \Lambda, 1/R$ then we can map the composites (see Table 1) onto the classical flat directions: $B_2 \sim A_{1,5}A_{2,4}q_{2,3}$, $\overline{B}_1 \sim A_{2,4}\overline{q}_{2,2}\overline{q}_{3,4}$, $M \sim \overline{q}_{1,1}q_{1,1}$, which is a baryonic-mesonic mixed branch. First we turn on only VEVs for $B_2$ (i.e. $a = b$ and $q_{2,3}$). To be able to do so, we have to restrict the adjoint VEVs to satisfy $v_3 = 0$, $v_2 + v_4 = 0$ and $v_1 + v_5 = 0$, so that the adjoint VEV is inside the low-energy gauge group. Thus the adjoint VEV is in the Cartan of $Sp(4)$:

$$\phi = \mathrm{diag}(v_1, v_2, 0, -v_2, -v_1) . \tag{5.12}$$

The gauge symmetry breaks[3] at the scale of the matter VEVs from $SU(5)$ to $Sp(4)$, and the unbroken Cartan elements are:

$$Q_{2+3} = \frac{1}{2}\,\mathrm{diag}(0, 1, 0, -1, 0) , \tag{5.13}$$

$$Q_{1+4} = \frac{1}{2}\,\mathrm{diag}(1, -1, 0, 1, -1) . \tag{5.14}$$

The broken $U(1)$ generators are:

$$Q_{2-3} = \frac{1}{2}\,\mathrm{diag}(0, 1, -2, 1, 0) , \tag{5.15}$$

$$Q_{1-4} = \frac{1}{2}\,\mathrm{diag}(1, -1, 0, -1, 1) . \tag{5.16}$$

The embedding of representations is shown in Table 4.

| $SU(5)$ | $\rightarrow$ | $SU(4)$ | $\rightarrow$ | $Sp(4)$ | $\rightarrow$ | $SU(2)$ |
|---|---|---|---|---|---|---|
| 5 | $\rightarrow$ | 4+1 | $\rightarrow$ | 4 +1 | $\rightarrow$ | (2+1+1)+1 |
| 10 | $\rightarrow$ | 6+4 | $\rightarrow$ | 5+1+4 | $\rightarrow$ | (2+2+1)+1+(2+1+1) |
| 24 | $\rightarrow$ | 15+4+$\overline{4}$+1 | $\rightarrow$ | 10+5+2·4 +1 | $\rightarrow$ | (3+2+2+3·1)+(2+2+1)+2(2+1+1)+1 |

Table 4: Embedding of representations of various subgroups into representations of $SU(5)$.

The vector supermultiplet eats the non-singlet pieces of the antisymmetric and one fundamental via the super-Higgs mechanism, so the low-energy $Sp(4)$ theory has four fundamentals (aka two flavors). The scales are related by

$$\Lambda^{11} = \Lambda^7_{(Sp)}\,ab^2\,q_{2,3} . \tag{5.17}$$

There are two confined composite monopoles that have the magnetic charges of the monopoles of the effective $Sp(4)$ theory and that are neutral under the broken $U(1)$'s, (5.15) and (5.16). One of them is comprised of monopoles 2 and 3, and the other is comprised of monopoles 1 and

---

[3]The description in terms of an $Sp(4)$ gauge group for $F = 2$ is only approximate since small gauge invariants (i.e. $\overline{B}_1$ and $M$ in this case) cannot be exactly zero given the superpotential (5.10).

4. Four of the ten zero modes of the 2+3 composite are lifted by an antisymmetric VEV and a fundamental VEV leaving six unlifted zero modes, so the 2+3 composite monopole, $Y_1^{(Sp)} = Y_2 Y_3 b\, q_{2,3} = Y_2 Y_3 A Q$, cannot contribute to the superpotential. Four of the six zero modes of the 1+4 composite are lifted by two antisymmetric VEVs leaving exactly two unlifted zero modes, so the 1+4 composite monopole, $Y_2^{(Sp)} = Y_1 Y_4 ab = Y_1 Y_4 A^2$, does contribute to the superpotential. With no further VEVs the superpotential would be

$$W = \eta_{(Sp)} Y_1^{(Sp)} Y_2^{(Sp)} + \frac{1}{Y_2^{(Sp)}} = \eta Y + \frac{1}{Y_1 Y_4\, A^2} \ . \tag{5.18}$$

A sketch of two composite monopoles with their zero modes is shown in Fig. 4. The structure of

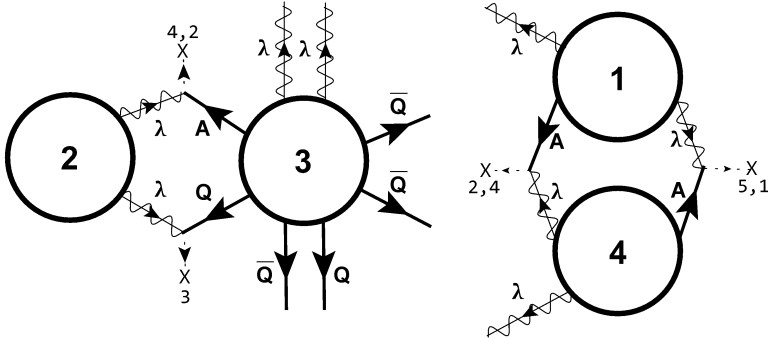

Figure 4: A sketch of the multi-monopole composites formed for $F = 2$ when the gauge group breaks from $SU(5)$ to $Sp(4)$. The antisymmetric and squark VEVs, (5.6) and (5.7), are represented as a cross with their color indices.

the low-energy effective theories and the resulting composite monopoles can be summarized in the affine (extended) Dynkin diagrams for $SU(5)$ and its $Sp(4)$ subgroup as shown in Fig. 5.

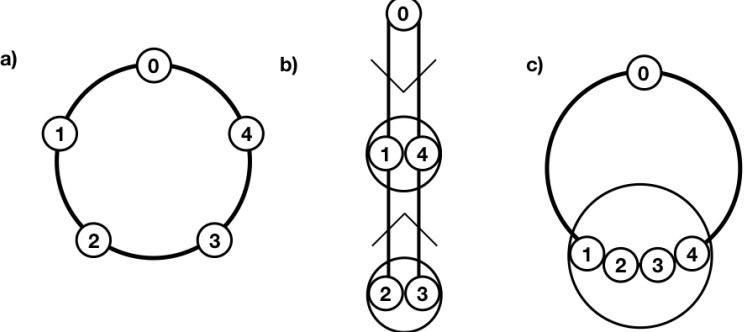

Figure 5: The extended Dynkin diagrams for breaking patterns of $SU(5)$: a) $SU(5)$, b) $Sp(4)$, and c) $SU(2)_a$. The directed double lines for $Sp(4)$ indicate that the roots are separated by $135°$, and the arrows point from a long root to a short root.

Turning on the $\overline{B}_1$ VEV (i.e the VEVs of $\overline{q}_{2,2}$ and $\overline{q}_{3,4}$) breaks $Sp(4)$ to $SU(2)_a$, and we have to restrict the adjoint VEV to the adjoint of the unbroken $SU(2)_a$, which means $v_2 \to 0$ and $v_4 \to 0$. Accounting for eaten Goldstone bosons and their superpartners leaves the effective $SU(2)_a$ theory

with one fundamental and one antifundamental. The unbroken Cartan generator is $Q_a$, given in Eq. (4.12), and the generator (5.13) is now broken. The scales of the gauge groups are related by

$$\Lambda^7_{(Sp)} = \Lambda^5_{(2a)} \, \bar{q}_{2,2} \, \bar{q}_{3,4} \; . \tag{5.19}$$

The 2+3 composite and 1+4 composite are now confined by the antifundamental VEVs and there is a composite comprised of monopoles 1, 2, 3 and 4 leaving four unlifted zero modes, so it cannot contribute to the superpotential. The superpotential comes entirely from the KK monopole:

$$W = \eta_{(2a)} Y^{(2a)} = \eta_{(Sp)} Y^{(Sp)}_1 Y^{(Sp)}_2 = \eta Y \; , \tag{5.20}$$

where $Y^{(2a)} = Y_1 Y_2 Y_3 Y_4 \, ab^2 \, q_{2,3} \, \bar{q}_{2,2} \, \bar{q}_{3,4} = Y_1 Y_2 Y_3 Y_4 \, A^2 Q \, A\overline{Q}^2$. A sketch of the 1+2+3+4 composite monopole is shown in Fig. 6.

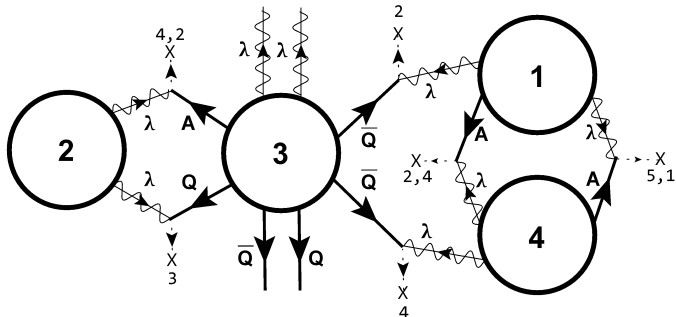

Figure 6: A sketch of the multi-monopole composite formed for $F = 2$ when the gauge group breaks from $SU(5)$ to $SU(2)_a$; the antisymmetric and squark VEVs, (5.6) and (5.7), are represented as a cross with their color indices.

Finally turning on the meson VEV $M$ (i.e. $q_{1,1} \neq 0$ and $\bar{q}_{1,1} \neq 0$) results in an instanton that is a confined composite of the KK monopole and the 1+2+3+4 composite, leaving only two unlifted zero modes. A sketch of the 1+2+3+4-KK instanton is shown in Fig. 7. The final superpotential

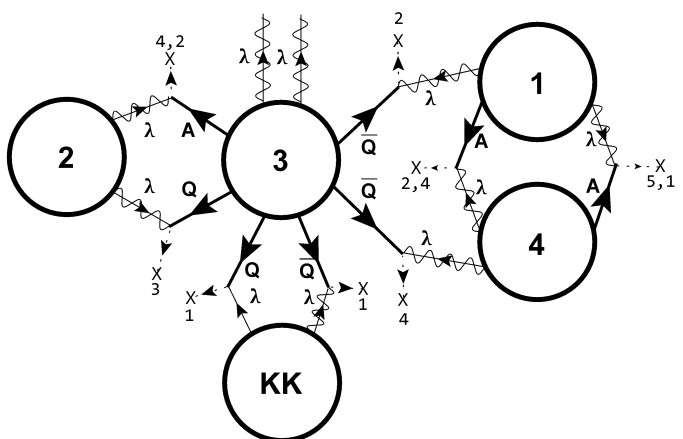

Figure 7: A sketch of the instanton for $F = 2$.

(which agrees with the 4D calculation) is:

$$W_{F=2} = \frac{\eta Y}{((Y_1 Y_4 ab)\,(Y_2 Y_3 \, b \, q_{2,3})\,\bar{q}_{2,2}\,\bar{q}_{3,4})\,q_{1,1}\,\bar{q}_{1,1}} = \frac{\eta}{B_2 \overline{B}_1 M} \; . \tag{5.21}$$

We can find the 3D limit by adding a real mass term for one flavor in the s-confining $\boxplus+3\square+4\overline{\square}$ theory. After integrating out the heavy flavor we obtain a low-energy 3D theory with a quantum modified constraint given by the superpotential

$$W_{3D,F=2} = \lambda \left( Y(B_2\overline{B}_1 M) - 1 \right). \tag{5.22}$$

In the 3D limit the zero-modes of the massive flavor jump [26] to the KK monopole and it decouples, leaving a composite monopole with no fermion zero modes. As with an instanton in 4D [28] this gives a contribution to a scalar n-point function, which (using cluster decomposition for gauge invariant operators) is equivalent to the constraint of the deformed moduli space (5.22). The monopole diagram, shown in Fig. 8, contributes to a 3-point-function of gauge invariants , in agreement with (5.22).

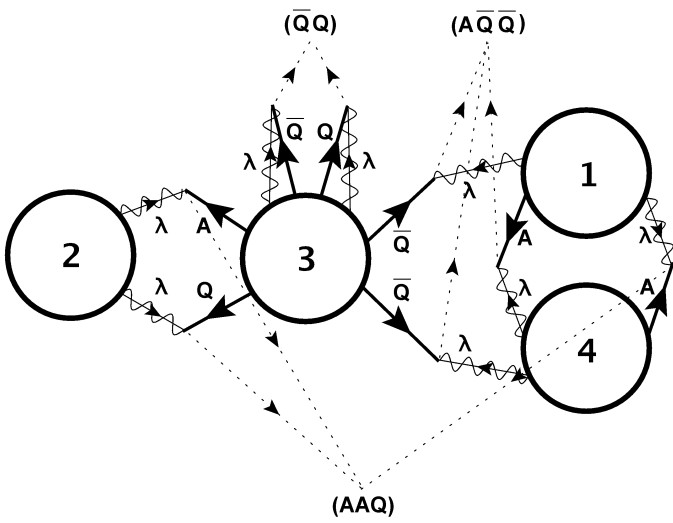

Figure 8: A sketch of the instanton contribution to the three-point function of gauge invariants for $F=2$ in the 3D, $R \to 0$ limit.

### 5.2.2  $F=2$, $B_2 \gg M \gg \overline{B}_1$

The case with hierarchical VEVs,

$$B_2 \gg M \gg \overline{B}_1 , \tag{5.23}$$

is similar to the previous subsection. For this case we use the squark VEVs

$$Q_{f\alpha} = \begin{pmatrix} 0 & 0 \\ q_{1,2} & 0 \\ 0 & q_{2,3} \\ 0 & 0 \\ 0 & 0 \end{pmatrix} , \quad \overline{Q}^*_{f,\alpha} = \begin{pmatrix} \overline{q}^*_{1,1} & 0 & 0 \\ 0 & \overline{q}^*_{2,2} & 0 \\ 0 & 0 & 0 \\ 0 & 0 & 0 \\ 0 & 0 & \overline{q}^*_{3,5} \end{pmatrix} , \tag{5.24}$$

where $D$-flatness requires

$$2|b|^2 = 2|b|^2 + |q_{1,2}|^2 - |\overline{q}_{2,2}|^2 = |q_{2,3}|^2 = 2|a|^2 - |\overline{q}_{1,1}|^2 = 2|a|^2 - |\overline{q}_{3,5}|^2 . \tag{5.25}$$

For large matter VEVs we can map the composites (see Table 1) onto the classical flat directions: $B_2 \sim A_{1,5}A_{2,4}q_{2,3}$, $M \sim \overline{q}_{2,2}q_{1,2}$, $\overline{B}_1 \sim A_{1,5}\overline{q}_{1,1}\overline{q}_{3,5}$. By turning on VEVs hierarchically, the gauge symmetry breaks from $SU(5) \rightarrow Sp(4) \rightarrow SU(2)_a$, and finally is broken completely. We again arrive the superpotential (5.21). In Appendix D we describe other regions of the lifted moduli space with different patterns of hierarchical VEVs. All cases reproduce the same superpotential, as expected.

## 5.3 $SU(5)$ with $F = 1$: $\boxvoid + \square + 2\overline{\square}$

We will start by looking at the parameterization of the antisymmetric VEV given in Eq. (5.6); $D$-flatness requires squark VEVs

$$
Q_\alpha = \begin{pmatrix} 0 \\ 0 \\ q_3 \\ 0 \\ 0 \end{pmatrix} , \quad \overline{Q}^*_{f,\alpha} = \begin{pmatrix} 0 & 0 \\ \overline{q}^*_{1,2} & 0 \\ 0 & 0 \\ 0 & \overline{q}^*_{2,4} \\ 0 & 0 \end{pmatrix} ,
\tag{5.26}
$$

that satisfy

$$
2|a|^2 = 2|b|^2 - |\overline{q}_{1,2}|^2 = |q_3|^2 = 2|b|^2 - |\overline{q}_{2,4}|^2 .
\tag{5.27}
$$

At a generic point on the moduli space the gauge group is completely broken and the moduli space is parameterized by gauge invariant composite mesons and baryons:

|  | $SU(2)$ | $U(1)_1$ | $U(1)_2$ | $U(1)_R$ |
|---|---|---|---|---|
| $M = Q\overline{Q}$ | $\square$ | -1 | -6 | -1 |
| $B_2 = A^2Q$ | 1 | -2 | 3 | 2 |
| $\overline{B}_1 = A\overline{Q}^2$ | 1 | 2 | -3 | -6 |

Table 5: Global quantum numbers of gauge invariant composite fields for $F = 1$.

The antisymmetric and squark VEVs, (5.6) and (5.26), are both invariant under $SU(2)_a$. For large VEVs we can map the composites (see Table 5) onto the classical flat directions:

$$
B_2 \sim A_{1,5}A_{2,4}q_3 \sim abq_3 ,
\tag{5.28}
$$
$$
\overline{B}_1 \sim A_{2,4}\overline{q}_{1,2}\overline{q}_{2,4} \sim b\overline{q}_{1,2}\overline{q}_{2,4} ,
\tag{5.29}
$$

and we see that our choice of parameterization has placed us on a baryonic branch with $M = 0$. We will also see shortly that there are (classically) also two meson branches with $M \neq 0$, one with $B_2 = 0$ and one with $\overline{B}_1 = 0$

At the point on the baryon branch described by (5.6) and (5.26) the adjoint VEV is restricted to

$$
\phi = \text{diag}(v_1, 0, 0, 0, -v_1) ,
\tag{5.30}
$$

and we see that this is a mixed Higgs-Coulomb branch. The VEVs of $A, Q, \overline{Q}$ break the gauge symmetry to $SU(2)_a$ and the low-energy theory has no flavors. The adjoint VEV (5.30) breaks $SU(2)_a$

down to $U(1)$, and there is a corresponding composite monopole, where 6 gaugino zero modes are lifted by three $A$ VEVs, one $Q$ VEV, and two $\overline{Q}$ VEVs. The corresponding superpotential is:

$$W_{F=1} = \eta Y + \frac{1}{Y_1 Y_2 Y_3 Y_4 B_2 \overline{B}_1} \ . \tag{5.31}$$

This corresponds to gaugino condensation in the low-energy $SU(2)$ gauge group (which is only broken by the adjoint VEV). In the 3D limit we find

$$W_{\text{3D},F=1,\text{baryonic}} = \frac{1}{Y_1 Y_2 Y_3 Y_4 B_2 \overline{B}_1} \ , \tag{5.32}$$

which has a runaway vacuum. Integrating out $Y = Y_1 Y_2 Y_3 Y_4$ from (5.31) we find the 4D superpotential:

$$W_{\text{4D},F=1,\text{baryonic}} = 2\epsilon \frac{\Lambda^6}{\sqrt{B_2 \overline{B}_1}} \ , \tag{5.33}$$

where $\epsilon = \pm 1$, so there are actually two runaway baryonic branches. Eq. (5.33) can also be derived from the $F = 2$ superpotential (5.10) by adding a quark mass for one of the flavor, i.e. adding $m M_{22}$ to the superpotential and integrating out the composites that contain the heavy flavor.

### 5.3.1 $F = 1$, Baryon Branch, $B_2 \gg \overline{B}_1$

We can first consider the hierarchical VEVs,

$$B_2 \gg \overline{B}_1 \ . \tag{5.34}$$

With a large VEV for $B_2$, the gauge symmetry breaks at the high scale from $SU(5)$ to $Sp(4)$, and the low-energy $Sp(4)$ theory[4] has two fundamentals. The scales are related by

$$\Lambda^{12} = \Lambda^8_{(Sp)} \, a b^2 \, q_3 \ . \tag{5.35}$$

The adjoint VEV has the form

$$\phi = \text{diag}(v_1, v_2, 0, -v_2, -v_1) \ . \tag{5.36}$$

The monopoles of the effective $Sp(4)$ theory are $Y_{\text{Sp},1} = Y_1 Y_4 \, a \, b$ and $Y_{\text{Sp},2} = Y_2 Y_3 \, b \, q_3$, where monopole 1 and 4 are confined, and monopole 2 and 3 are confined to be neutral under the broken generators. In addition to the standard two gaugino zero modes, $Y_{\text{Sp},2}$ has two extra zero modes corresponding to the two fundamentals. With no further VEVs the superpotential is

$$W = \eta_{(Sp)} Y_{\text{Sp},1} Y_{\text{Sp},2} + \frac{1}{Y_{\text{Sp},1}} = \eta Y + \frac{1}{Y_1 Y_4 \, ab} \ . \tag{5.37}$$

Sketches of the two composite monopoles are shown in Fig. 9.

Turning on the VEVs for the two fundamentals of the effective $Sp(4)$ theory breaks $Sp(4)$ to $SU(2)_a$, and the low-energy effective theory has no matter fields. The scales are related by

$$\Lambda^8_{(Sp)} = \Lambda^6_{(2)} \, \overline{q}_{1,2} \overline{q}_{2,4} \ . \tag{5.38}$$

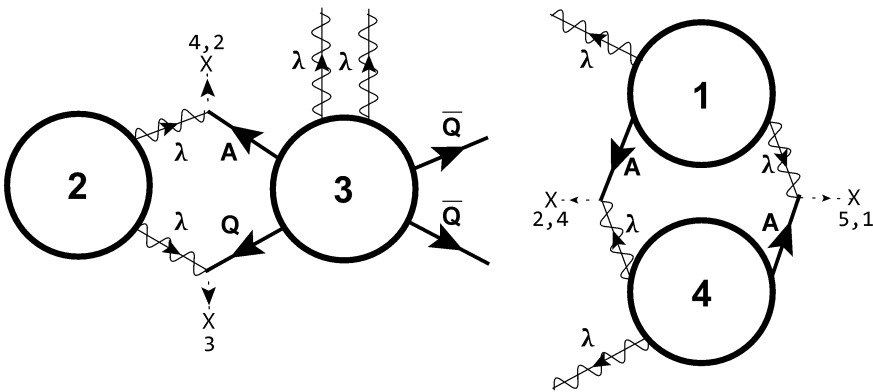

Figure 9: The multi-monopole composites for $F = 1$ on the baryonic branch when the gauge group breaks from $SU(5)$ to $Sp(4)$. The antisymmetric and squark VEVs, (5.6) and (5.26), are represented as a cross with their color indices.

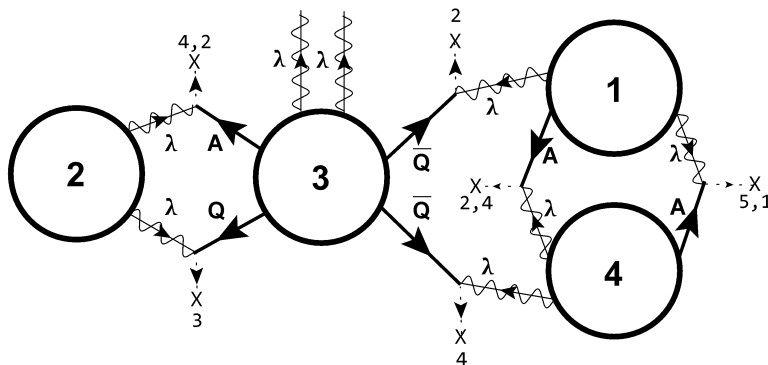

Figure 10: The multi-monopole composite contributing to the superpotential for $F = 1$ when the gauge group breaks from $SU(5)$ to $SU(2)_a$. The antisymmetric and squark VEVs, (5.6) and (5.26), are represented as a cross with their color indices.

The composite monopoles $Y_{\text{Sp},1}$ and $Y_{\text{Sp},2}$ are now confined and make a composite comprised of monopoles 1+2+3+4, leaving two unlifted zero modes, so this multi-monopole contributes to the superpotential. A sketch of the 1+2+3+4 composite monopole is shown in Fig. 10.

The superpotential is

$$W_{F=1,\text{baryonic}} \quad = \quad \eta_2 Y_{SU(2)} + \frac{1}{Y_{SU(2)}} = \eta_{(Sp)} Y_{\text{Sp},1} Y_{\text{Sp},2} + \frac{1}{Y_{\text{Sp},1} Y_{\text{Sp},2} \, \overline{q}_{1,2} \overline{q}_{2,4}} \tag{5.39}$$

$$= \quad \eta Y + \frac{1}{Y a\, b^2\, q_3\, \overline{q}_{1,2} \overline{q}_{2,4}} \tag{5.40}$$

$$= \quad \eta Y + \frac{1}{Y B_2 \overline{B}_1} \;. \tag{5.41}$$

Integrating out the Coulomb branch moduli $Y$ we recover (5.33).

---

[4]The $Sp(4)$ gauge symmetry is broken further unless $\overline{B}_1$ is exactly zero.

### 5.3.2 $F = 1$, Baryon Branch, $\overline{B}_1 \gg B_2$

Next, let's consider the case with hierarchical VEVs,

$$\overline{B}_1 \gg B_2 \ . \tag{5.42}$$

With a $\overline{B}_1$ VEV turned on, we have a large antisymmetric VEV $b$ as well as VEVs for $\overline{q}_{1,2}$ and $\overline{q}_{2,4}$. The $b$ VEV is invariant under $SU(3)_a \times SU(2)_b$. The $\overline{q}_{1,2}$ and $\overline{q}_{2,4}$ VEVs further reduce the gauge symmetry to $SU(3)_a$, and the low-energy theory has one fundamental and one antifundamental (one flavor). There confined monopoles are 1+2 and 3+4, while the KK monopole is neutral under the broken generators. The scales of the $SU(5)$ theory and the low-energy $SU(3)_a$ theory are related by

$$\Lambda^{12} = \Lambda_{(3)}^8 \, b^2 \overline{q}_{1,2} \, \overline{q}_{2,4} \ . \tag{5.43}$$

In region (4.9) of the fundamental Weyl chamber, turning on a VEV for $\overline{B}_1$ further restricts the adjoint VEV $v_{2,3,4} \to 0$ and $v_1 + v_5 \to 0$, as in (5.30). Thus the VEVs for $B_2$ (i.e. $a$ and $q_3$) can be turned on without any further restriction on the adjoint. The VEVs $a$ and $q_3$ break $SU(3)_a$ to $SU(2)_a$ leaving no matter fields in the effective theory. The scales are related by

$$\Lambda_{(3)}^8 = \Lambda_{(2)}^6 \, a \, q_{2,3} \ . \tag{5.44}$$

The neutral composite monopole is again 1+2+3+4, as shown in Fig. 10. The 1+2+3+4 monopole has two unlifted gaugino zero modes, so it contributes to the superpotential. We again arrive the superpotential (5.41), and by integrating out the Coulomb branch moduli we recover (5.33).

In Appendix. C we discuss composite monopoles that appear in another region of the Weyl chamber region of Table 8 which allows an adjoint under the $SU(3)$ on its boundary.

### 5.3.3 $F = 1$, Lifted Meson Branch

We can also consider the case of the meson branch where $M$ is the largest gauge invariant VEV. It will be useful to consider the following field VEVs:

$$A = \begin{pmatrix} 0 & 0 & 0 & 0 & a \\ 0 & 0 & 0 & a & 0 \\ 0 & 0 & 0 & b & 0 \\ 0 & -a & -b & 0 & 0 \\ -a & 0 & 0 & 0 & 0 \end{pmatrix} \ , \tag{5.45}$$

$$Q_\alpha = \begin{pmatrix} 0 \\ 0 \\ q_3 \\ 0 \\ 0 \end{pmatrix} \ , \quad \overline{Q}_{f,\alpha}^* = \begin{pmatrix} 0 & 0 \\ 0 & 0 \\ \overline{q}_{1,3}^* & 0 \\ 0 & \overline{q}_{2,4}^* \\ 0 & 0 \end{pmatrix} \ . \tag{5.46}$$

$D$-flatness requires

$$2|a|^2 = 2|a|^2 = 2|b|^2 + |q_3|^2 - |\overline{q}_{1,3}|^2 = 2|a|^2 + 2|b|^2 - |\overline{q}_{2,4}|^2 \ , \tag{5.47}$$

and

$$2ab = 0 \; . \tag{5.48}$$

For large matter VEVs we can map the composites (see Table 5) onto the classical flat directions:

$$M_1 \quad \sim \quad \overline{q}_{1,3}\, q_3 \tag{5.49}$$

$$\overline{B}_1 \quad \sim \quad A_{3,4}\overline{q}_{1,3}\overline{q}_{2,4} \sim b\,\overline{q}_{1,3}\overline{q}_{2,4} \tag{5.50}$$

$$B_2 \quad \sim \quad A_{1,5}A_{2,4}q_3 \sim a^2\, q_3 \; . \tag{5.51}$$

So we see that classically there are two meson branches: one where $B_2$ vanishes and one where $\overline{B}_1$ vanishes, depending on whether we choose $a = 0$ or $b = 0$. Since the baryon branch already gives all the branches obtained by integrating out one flavor from the $F = 2$ case (5.10) we should expect that the meson branches are completely lifted, the question is: how are the meson branches lifted?

Turning on the VEVs $q_3$ and $\overline{q}_{1,3}$ (i.e. an $M$ VEV) breaks the gauge symmetry from $SU(5)$ to $SU(4)$, and the low-energy $SU(4)$ theory has an antisymmetric, one fundamental and one antifundamental; the fundamental comes from the components of antisymmetric tensor $A$. The scales are related by

$$\Lambda^{12} = \Lambda_{(4)}^{10}\, q_3\, \overline{q}_{1,3} \; . \tag{5.52}$$

Turning on the $a$ VEV (i.e. a $B_2$ VEV), breaks the gauge symmetry from $SU(4)$ to $Sp(4)$, and the low-energy $Sp(4)$ theory has two fundamentals. The scales are related by

$$\Lambda_{(4)}^{10} = \Lambda_{(Sp)}^{8}\, a^2 \; . \tag{5.53}$$

The composite monopoles are 2+3 and 1+4 as shown in Fig. 9. Since the low-energy $Sp(4)$ theory has two fundamentals there is a low-energy $D$-flat direction which would run away, at least according to the ADS superpotential. This flat direction corresponds to $b\,\overline{q}_{2,4} \neq 0$, however we know that this is not a $D$-flat direction of the full $SU(5)$ theory. This is an example of a non-decoupling $D$-term [29]. Without the non-decoupling $D$-term the gauge group breaks to $SU(2)$, the scales would be related by

$$\Lambda_{(Sp)}^{8} = \Lambda_{(2)}^{6}\, b\,\overline{q}_{2,4} \; , \tag{5.54}$$

and the superpotential would be

$$W_{F=1,\mathrm{SU}(2)} \quad = \quad \eta Y + \frac{1}{Y\,\overline{q}_{1,3}\, q_3\, a^2\, b\,\overline{q}_{2,4}} \; . \tag{5.55}$$

The full $D$-term potential is

$$D^a \quad = \quad -T_n^{am}\left(\langle A^\dagger A\rangle + \langle AA^\dagger\rangle + \langle Q^\dagger Q\rangle - \langle \overline{Q}^\dagger \overline{Q}\rangle\right)_m^n \tag{5.56}$$

$$V_D \quad = \quad \frac{1}{2}D^a D^a = \frac{1}{10}\left(8|a|^4 + 12|b|^4 + |a|^2\left(8|b|^2 - 8|q_3|^2 + 8|\overline{q}_{1,3}|^2 - 2|\overline{q}_{2,4}|^2\right)\right.$$
$$\left. +6|b|^2\left(|q_3|^2 - |\overline{q}_{1,3}|^2 - |\overline{q}_{2,4}|^2\right) + 2|q_3|^4 - 4|q_3|^2|\overline{q}_{1,3}|^2 \right.$$
$$\left. +|q_3|^2|\overline{q}_{2,4}|^2 + 2|\overline{q}_{1,3}|^4 - |\overline{q}_{1,3}|^2|\overline{q}_{2,4}|^2 + 2|\overline{q}_{2,4}|^4\right) \; , \tag{5.57}$$

where $T^a$ are the $SU(5)$ generators.

Let's suppose $b, \overline{q}_{2,4} \ll a$ (so that the SUSY breaking is parametrically small). In this limit we can minimize the full scalar potential:

$$
\begin{aligned}
V &= \left|\frac{\partial W}{\partial a}\right|^2 + \left|\frac{\partial W}{\partial b}\right|^2 + \left|\frac{\partial W}{\partial q_3}\right|^2 + \sum_i \left|\frac{\partial W}{\partial \overline{q}_i}\right|^2 + \left|\frac{\partial W}{\partial Y}\right|^2 + V_D \tag{5.58} \\
&= \frac{1}{|a^2\, b\, q_3\, \overline{q}_{1,3}\, \overline{q}_{2,4}\, Y|^2}\left(\frac{4}{|a|^2} + \frac{1}{|b|^2} + \frac{1}{|q_3|^2} + \frac{1}{|\overline{q}_{1,3}|^2} + \frac{1}{|\overline{q}_{2,4}|^2} + \frac{1}{|Y|^2}\right) \\
&\quad + |\eta|^2 - 2\mathrm{Re}\left[\frac{\eta}{a^2\, b\, q_3\, \overline{q}_{1,3}\, \overline{q}_{2,4}\, Y^2}\right] + V_D\ , \tag{5.59}
\end{aligned}
$$

Since $Y$ is bounded on $R^3 \times S^1$ we see that the semi-classical $F$-terms diverge if any of the matter VEVs goes to zero, while the $D$-term potential is only minimized at $b = \overline{q}_{2,4} = 0$, so SUSY is broken on this branch, or in other words, this branch is lifted. The potential is minimized with respect to $Y$ by:

$$
Y = \pm\sqrt{\frac{1}{\eta\, a^2\, b\, q_3\, \overline{q}_{1,3}\, \overline{q}_{2,4}}}\ . \tag{5.60}
$$

Since $Y$ is dimensionless we can easily restore the dependence on $R$ using $\eta = (R\Lambda)^{12}$, which gives

$$
Y = \pm\sqrt{\frac{1}{R^{18}\, \Lambda^{12}\, a^2\, b\, q_3\, \overline{q}_{1,3}\, \overline{q}_{2,4}}}\ . \tag{5.61}
$$

Since $V$ is dimension 4, we have

$$
V = \frac{\Lambda^{12}}{|a^2\, b\, q_3\, \overline{q}_{1,3}\, \overline{q}_{2,4}|}\left(\frac{4}{|a|^2} + \frac{1}{|b|^2} + \frac{1}{|q_3|^2} + \frac{1}{|\overline{q}_{1,3}|^2} + \frac{1}{|\overline{q}_{2,4}|^2}\right) + V_D\ . \tag{5.62}
$$

For $a \simeq q_3 \simeq \overline{q}_{1,3}$ and $b \simeq \overline{q}_{2,4}$ the potential is minimized with

$$
b,\ \overline{q}_{2,4} \propto \frac{\Lambda^2}{a}\ , \tag{5.63}
$$

so for $a \gg \Lambda$, SUSY breaking is indeed parametrically small, as we assumed. More generally, as we will see in Sec. 6, even without supersymmetry, monopoles contribute to the scalar potential with inverse powers of the scalar VEVs, so with the $D$-term potential (5.57) SUSY must be broken on this branch.

Alternatively taking $b \sim \overline{q}_{2,4} \gg a$ we break from $SU(4)$ to a low-energy $SU(3)$ theory with a fundamental and an antifundamental. The low-energy $SU(3)$ theory by itself has a $D$-flat direction that corresponds to the $a$ direction, but with $b \neq 0$ this is not a flat direction of $SU(5)$. Again the $F$-terms diverge at $a = 0$ while the $D$-terms are minimized at $a = 0$. So we have shown that in either case, $\overline{B}_1 = 0$ or $B_2 = 0$, the meson branch is completely lifted, as expected.

For both cases, in the 3D limit, $Y$ is no longer bounded and the matter VEVs can approach zero as $Y$ runs away to infinity on the Coulomb branch.

# 6   SUSY Breaking: $SU(5)$ with ⊟ $+ \bar{\Box}$

It is well known that this theory breaks SUSY, this has been argued from a variety of perspectives [30–40]. From our results in section 5 we can see a simple new argument for SUSY breaking. Adding a mass term for the flavor in the $F = 1$ theory to the superpotential (5.33) we have

$$W_{\text{break}} = 2\epsilon \frac{\Lambda^6}{\sqrt{B_2 \overline{B}_1}} + m M_{11} \ . \tag{6.1}$$

Since the ADS superpotential term is independent of the meson we see that this breaks SUSY by the Polonyi mechanism [41]. For a more dynamical understanding we can return to $R^3 \times S^1$, but first it is worth noting what we should expect to find. There are no $D$-flat directions, so there is no moduli space. There are however two gauge invariant operators composed of $\overline{Q}$, $A$ and gauginos:

$$
\begin{aligned}
S &= \lambda^s_{c_1} \lambda^r_s \overline{Q}^t A_{tr} A_{c_2 c_3} A_{c_4 c_5} \varepsilon^{c_1 c_2 c_3 c_4 c_5} \ , \tag{6.2} \\
S' &= \lambda^s_a \lambda_{s,bc_1 dc_2} \overline{Q}^r A_{tu} A_{rc_3} A_{c_4 c_5} \varepsilon^{abdtu} \varepsilon^{c_1 c_2 c_3 c_4 c_5} \ , \tag{6.3}
\end{aligned}
$$

The tensor products of representations of $SU(5)$ corresponding to the gauge singlets $S$ and $S'$ are shown in Table 6.

$$
\begin{aligned}
(24 \times 24)_{\text{s}} &= 1 + 24 + 75 + 200 \\
\bar{5} \times 10 &= 5 + 45 \\
10 \times 10 &= \bar{5}_{\text{s}} + \overline{45}_{\text{a}} + \overline{50}_{\text{s}} \\
(10 \times 10 \times 10)_{\text{s}} &= 45 + 45 + 45 + 175'' \\
\bar{5} \times 45 &= 24 + 75 + \overline{126}
\end{aligned}
$$

Table 6: The tensor products of representations of $SU(5)$.

The difference between $S$ and $S'$ is just whether the matter fields are contracted to the 24 or the 75 dimensional representation. Instanton calculations require that at least one of $S$ or $S'$ must be non-zero [30–32, 34, 36]. Taking the instanton generated 't Hooft vertex and connecting all the matter fermion zero-modes to a gaugino zero mode using a Yukawa coupling to a scalar we see that the instanton gives a non-zero amplitude for $S$ or $S'$ as well as two gauge invariants formed from gaugino bilinears. A non-zero VEV for $S$ or $S'$ requires that the gauge symmetry is broken to $SU(2)$. In our standard region of the Weyl chamber (4.9) this we are restricted to $SU(2)_a$. The embedding of representations is shown in Table 7.

$$
\begin{aligned}
SU(5) &\rightarrow SU(2) \\
5 &\rightarrow \mathbf{2} + 3 \cdot \mathbf{1} \\
10 &\rightarrow 3 \cdot \mathbf{2} + 4 \cdot \mathbf{1} \\
24 &\rightarrow \mathbf{3} + 6 \cdot \mathbf{2} + 9 \cdot \mathbf{1}
\end{aligned}
$$

Table 7: Embedding of representations $SU(2)_a$ into representations of $SU(5)$.

We can examine the SUSY breaking with VEVs for the scalar components of $A$ and $\overline{Q}$ given

by:

$$
A = \begin{pmatrix} 0 & 0 & 0 & 0 & a \\ 0 & 0 & 0 & b & 0 \\ 0 & 0 & 0 & 0 & 0 \\ 0 & -b & 0 & 0 & 0 \\ -a & 0 & 0 & 0 & 0 \end{pmatrix} , \quad \overline{Q}_\alpha^* = \begin{pmatrix} 0 \\ 0 \\ 0 \\ \overline{q}^* \\ 0 \end{pmatrix} , \tag{6.4}
$$

which have a non-vanishing $D$-term potential, and break the gauge symmetry from $SU(5)$ to $SU(2)_a$. The $D$-term potential is:

$$
D^a = -T_n^{am} \left( \langle A^\dagger A \rangle + \langle A A^\dagger \rangle - \langle \overline{Q}^\dagger \overline{Q} \rangle \right)_m^n \tag{6.5}
$$

$$
V_{D-\text{term}} = \frac{1}{2} D^a D^a = \frac{1}{5} \left( 6|a|^4 + 6|b|^4 + |\overline{q}|^4 - 8|b|^2|a|^2 + 2|a|^2|\overline{q}|^2 - 3|b|^2|\overline{q}|^2 \right) . \tag{6.6}
$$

Note that there are not enough massless bosons for the super Higgs mechanism to occur, since each vector supermultiplet would have to eat an entire chiral supermultiplet. However without SUSY each massive gauge boson needs to eat only one real scalar degree of freedom and there are enough Goldstone bosons for this non-SUSY breaking pattern. So even before accounting for the $D$-terms, SUSY must be broken in order to reduce the unbroken gauge symmetry down to $SU(2)_a$. This also means that there are some gauginos that remain massless even though their superpartner gauge bosons become massive. This is analogous to what happens in SUSY QCD with a boundary condition that forces a VEV for a fundamental but not for an antifundamental. In our case, massless broken gauginos appear in both doublet and singlet representations of $SU(2)_a$.

With matter VEVs breaking $SU(5)$ down to $SU(2)_a$, the unbroken Cartan element is $Q_{1+2+3+4}$ as given in (4.12). The monopoles must be confined to form a composite monopole as shown in Fig. 11a.

This gauge breaking pattern produces a variety of elementary and composite monopoles. Since the VEVS do not allow for a supersymmetric spectrum, we cannot discuss their effects using a superpotential. However, as described in ref. [20], we can look at the scalar potential terms generated by joining monopoles to antimonopoles by connecting all the unlifted gaugino legs with ordinary propagators. First let us look at the various types of 't Hooft vertices that are produced. The simplest vertex is just from the KK monopole which (as we saw in section 4) can form bound states but is not confined. The corresponding 't Hooft vertex is:

$$
\mathcal{O}_{\text{KK}} = R^{12} \Lambda^{13} Y \lambda^2 , \tag{6.7}
$$

where we have reintroduced the dependence on $R$ using $\eta = (R\Lambda)^{13}$. The monopole of $SU(2)_a$ shown in Fig. 11a generates an 't Hooft vertex given by

$$
\mathcal{O}_a = \frac{a^* b^{*2} \overline{q}^*}{R^2 |a|^2 |b|^2 (|b|^2 + |\overline{q}|^2)^2 Y} \lambda^4 . \tag{6.8}
$$

To understand the field dependence we need to recall the path-integral calculation of ref. [11], which included integrations over bosonic and fermionic zero modes. Three of the bosonic zero modes are collective coordinates representing the location of the center of the monopole in $R^3$. There are also collective coordinates for each of the flux-tube lengths, $\rho_i$. In this case we have reduced the rank of the gauge group by 3, so there are three collective coordinates corresponding to

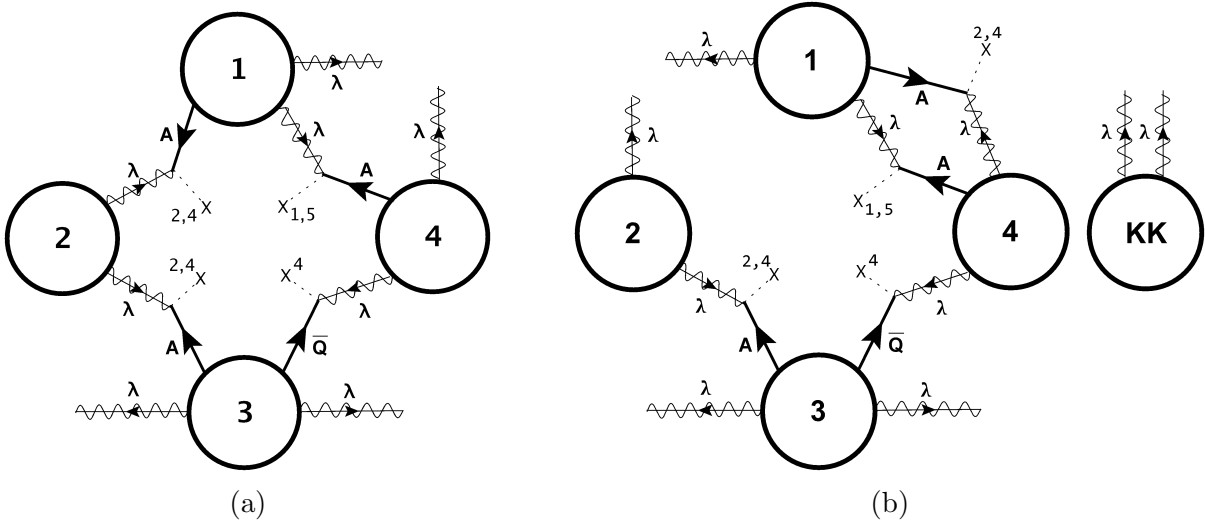

(a)                                                    (b)

Figure 11: A sketch of multi-monopole composites for $F = 0$. The monopole that generates the 't Hooft vertex (6.8) is shown in (a) and the composite that generates the 't Hooft vertex (6.10) is shown in (b). The antisymmetric and squark VEVs, (6.4), are represented as a cross with their color indices. Two of the gaugino zero modes in (b) remain massless even though their superpartner gauge bosons are massive. Note the the KK monopole can form a bound state with the 1-4 composite, as in Sec. 4.

relative monopole positions. When we integrate over these collective coordinates the exponential damping by the gauge boson mass term in the action gives a dominant contribution [11] from

$$\rho_i \sim \frac{1}{R|M_i|^2} \ , \tag{6.9}$$

where $M_i$ is the mass of the broken $U(1)$ gauge boson associated with the flux tube. In the case at hand the flux-tube between monopoles 1 and 4 is set by the VEV $a$, between 2 and 3 by the VEV $b$. However between the composite monopoles 1+4 and 2+3, both the $b$ and $\overline{q}$ VEVs contribute to the gauge boson mass, so we find the non-holomorphic structure in Eq. (6.8).

There is also an instanton generated vertex from the breaking of $SU(2)_b$ as shown in Fig. 11b which yields

$$\mathcal{O}_b = \frac{R^{14} \Lambda^{13} \, a^* b^{*2} \overline{q}^*}{(|a|^2 + |b|^2)^2 |b|^2 |\overline{q}|^2} \lambda^6 \ . \tag{6.10}$$

In this case the final breaking of $SU(2)_b$ is entirely due to the $\overline{q}$ VEV, so the mass scale for this collective coordinate is set entirely by this VEV, while the flux-tube between monopoles 1 and 4 depends on both $a$ and $b$.

The monopole–anti-monopole and instanton–anti-instanton contributions to the scalar potential are found by integrating out gaugino lines that leave the monopole/instanton and enter the anti-monopole/instanton [20]. There are further contributions where scalar VEVs are removed and the left-over legs are connected in the same fashion as the gaugino legs were. These latter contributions do not qualitatively change the results and, in any case, are less singular for small VEVs.

As in section 5.3.3 we find that since $Y$ is bounded there is a semi-classical contribution to the scalar potential that diverges when any of the VEVs vanish, while the $D$-term potential is only minimized when all the VEVs vanish, so SUSY is indeed broken.

In the 3D limit, $Y$ can become arbitrarily large, so the semi-classical potential terms can become arbitrarily small while the $D$-term potential is minimized with all the matter VEVs approaching zero, so we have a runaway vacuum on the Coulomb branch.

# 7 Conclusions

In this paper we have investigated $\mathcal{N} = 1$ supersymmetric chiral gauge theories compactified on $R^3 \times S^1$. Monopole confinement via rank reduction of the gauge group dynamically generates superpotentials which can be calculated semi-classically. We found that the structure of the composite multi-monopoles can be read off from the affine Dynkin diagrams of the gauge group and its unbroken subgroup. Taking the 4D limit by integrating out the Coulomb branch moduli results in ADS-like 4D superpotentials. The pre-ADS superpotentials on $R^3 \times S^1$ for matter content $\Box\!\Box + \overline{\overline{\Box}}$ and $\Box\!\Box + F\,\Box + (F+1)\,\overline{\Box}$ were studied in detail, and their 4D limits were found to be correct, which provides an important cross-check on the calculations. The $F = 1$ case is particularly interesting since we were able to show that the meson branch is completely lifted and that the superpotential only depends on the baryon composite fields. This in itself is enough to show, in a novel way, that SUSY is broken when the single flavor is integrated out. For the $F = 0$ case we were able to show that the composite monopoles drive SUSY breaking even though the analysis is much more complicated due to the manifest absence of supersymmetry in the spectrum.

# Acknowledgments

We thank, Csaba Csáki, Nick Dorey, Ryuichiro Kitano, Erich Poppitz, Nathan Seiberg, Yuri Shirman, David Tong, and Mithat Unsal for useful comments and discussions. This work was supported in part by the DOE under grant DE-SC-0009999.

# Appendix

# A    Zero Mode Conditions on Monopoles

The condition for a zero mode from the Callias index theorem [42–44] is that the absolute value of the singlet mass contribution, $|m|$, is smaller than the adjoint VEV contribution to the mass, $\frac{v}{2}$:

$$|m| < \frac{v}{2} \, , \tag{A.1}$$

where $v$ is the asymptotic adjoint scalar VEV. For an $SU(2)$ doublet from a fundamental representation of $SU(N)$ we have

$$\left| \frac{1}{2} \operatorname{Tr} v(\mathbf{h} - (\mathbf{h} \cdot \alpha_i)\alpha_i) \cdot \mathbf{H})P_i \right| < \frac{v(\mathbf{h} \cdot \alpha_i)}{2} \, , \tag{A.2}$$

where $P_i$ with $i = 1, 2, \ldots, N-1$ is a projector onto the $SU(2)$ subspace:

$$P_i = 4(\alpha_i \cdot \mathbf{H})^2 \, . \tag{A.3}$$

We can write the $SU(N)$ asymptotic adjoint VEV, up to a gauge transformation, as

$$\phi = \text{diag}(v_1, v_2, \cdots, v_N) , \qquad v_1 + v_2 + \cdots + v_N = 0 , \tag{A.4}$$

where

$$vh \cdot \alpha_i = v_i - v_{i+1} > 0 , \tag{A.5}$$

which requires that we are inside the fundamental Weyl chamber (3.16). Then the zero mode condition for the fundamental representation (A.2) on the $i$'th BPS monopole reads

$$|v_i + v_{i+1}| < v_i - v_{i+1} . \tag{A.6}$$

For the antisymmetric representation we need a little more work. We can decompose the representations of $SU(N)$ into representations of $SU(2) \times SU(N-2) \times U(1)$ as

$$\square = \mathbf{N} \to (\mathbf{2}, \mathbf{1})_{\frac{N-2}{\sqrt{2}N}} + (\mathbf{1}, \mathbf{N-2})_{\frac{-\sqrt{2}}{N}} , \tag{A.7}$$

$$\Box = \frac{\mathbf{N(N-1)}}{\mathbf{2}} \to (\mathbf{1}, \mathbf{1})_{\frac{\sqrt{2}(N-2)}{N}} + (\mathbf{1}, \frac{\mathbf{(N-2)(N-3)}}{\mathbf{2}})_{\frac{-2\sqrt{2}}{N}} + (\mathbf{2}, \mathbf{N-2})_{\frac{N-4}{\sqrt{2}N}} . \tag{A.8}$$

The antisymmetric representation decomposes into $N-2$ doublets under the $SU(2)$ subgroup and there is an additional singlet contribution to the mass.

The zero mode condition for a $k$'th doublet $(A_{i,k}, A_{i+1,k})$ of an antisymmetric tensor on the $i$'th BPS monopole is

$$\left| \frac{1}{2} \text{Tr}\, v(\mathbf{h} - (\mathbf{h} \cdot \alpha_i)\alpha_i) \cdot \mathbf{H})P_i + m_A \right| < \frac{v(\mathbf{h} \cdot \alpha_i)}{2} , \tag{A.9}$$

which in the fundamental Weyl chamber reads

$$\left| \frac{v_i + v_{i+1}}{2} + v_k \right| < \frac{v_i - v_{i+1}}{2} . \tag{A.10}$$

The condition (A.10) can be written explicitly as

$$\begin{cases} v_i > |v_k| > v_{i+1} > 0 > v_k \\ |v_{i+1}| > v_k > |v_i| > 0 > v_i > v_{i+1} \\ v_i > |v_k| > |v_{i+1}| > 0 > v_{i+1} > v_k \\ |v_{i+1}| > v_k > v_i > 0 > v_{i+1} . \end{cases} \tag{A.11}$$

It is possible for zero modes to exist on the KK monopole for $SU(N)$ when $N > 4$. Around the KK monopole we have an anti-periodic fermion solution with time dependence $\exp(\pm ix_4/2R)$. In 4D the $\partial_4$ derivative shifts the fermion mass by $\pm 1/2R$, which means the 3D Dirac equation has an effective real mass [26]

$$m_{\text{eff}} = m \mp \frac{1}{2R} . \tag{A.12}$$

The asymptotic adjoint VEV is also replaced as [23]

$$v' = \frac{1}{R} - v , \tag{A.13}$$

where $v$ is the asymptotic VEV of the adjoint under the $SU(2)$ subgroup corresponding to sum of the $N-1$ simple roots of $SU(N)$. Thus the zero mode condition (A.1) for the KK monopole is translated to $|m \mp \frac{1}{2R}| < \frac{1}{2R} - \frac{v}{2}$. In other words the zero mode condition is satisfied provided

$$m > \frac{v}{2} \quad \text{or} \quad m < -\frac{v}{2} \, , \tag{A.14}$$

which can be translated as

$$\left| \frac{1}{2} \operatorname{Tr} v(\mathbf{h} - (\mathbf{h} \cdot \alpha_0)\alpha_0) \cdot \mathbf{H})P_0 + m_r \right| > \frac{v(\mathbf{h} \cdot \alpha_i)}{2} \, , \tag{A.15}$$

where $\alpha_0 = \alpha_1 + \alpha_2 + \cdots + \alpha_{N-1}$ is the sum of the $N-1$ simple roots, $P_0$ is a projector onto the $SU(2)$ subspace corresponding to $\alpha_0$, and $m_r$ is an possible additional real mass contribution other than a real mass from adjoint VEVs of the $SU(2)$ subspace.

In the fundamental Weyl chamber the following inequality is always true:

$$\left| \frac{1}{2} \operatorname{Tr} v(\mathbf{h} - (\mathbf{h} \cdot \bar{\alpha})\bar{\alpha}) \cdot \mathbf{H})P_0 \right| = \frac{|v_1 + v_N|}{2} < \frac{v_1 - v_N}{2} = \frac{v(\mathbf{h} \cdot \bar{\alpha})}{2} \, . \tag{A.16}$$

Thus the zero mode condition on the KK monopole, (A.14), for the fundamental representation under $SU(N)$ cannot be satisfied unless there is an additional real mass contribution. For the antisymmetric representation the zero mode condition for a $k$'th doublet $(A_{1,k}, A_{N,k})$ on the KK monopole, i.e. (A.15), reads

$$\left| \frac{v_1 + v_N}{2} + v_k \right| > \frac{v_1 - v_N}{2} \, , \tag{A.17}$$

where $k = 2, 3, \ldots, N-1$. For $N \leq 4$, the condition (A.17) cannot be satisfied and there is no zero mode on the KK monopole. This can be proved as follows. We can first assume $v_1 > |v_N|$. Then for $N \leq 4$ it is easy to check that $|v_N| > v_k > v_N$ where the first inequality holds because otherwise $v_i$'s cannot sum up to zero, and the second inequality comes from the fundamental Weyl chamber condition (3.16). Then we get

$$\left| \frac{v_1 + v_N}{2} + v_k \right| < \left| \frac{v_1 + v_N}{2} \right| + |v_k| < \left| \frac{v_1 + v_N}{2} \right| + |v_N| = \frac{v_1 + v_N}{2} - v_N = \frac{v_1 - v_N}{2} \, , \tag{A.18}$$

so the condition (A.17) cannot be satisfied for $N \leq 4$. When $v_1 < |v_N|$ we analogously have $v_1 > |v_k|$ for the tracelessness and get

$$\left| \frac{v_1 + v_N}{2} + v_k \right| < \left| \frac{v_1 + v_N}{2} \right| + |v_k| < \left| \frac{v_1 + v_N}{2} \right| + |v_1| = -\frac{v_1 + v_N}{2} + v_1 = \frac{v_1 - v_N}{2} \, , \tag{A.19}$$

which concludes the proof. For $N > 4$ there can be a zero mode on the KK monopole in some region of the Weyl chamber. The $N = 5$ case is explicitly studied in next section.

# B   Coulomb Branches and Operators

On the circle, in the absence of matter, $SU(N)$ is broken down to $U(1)^{N-1}$ by the adjoint scalar giving a Coulomb branch moduli space. Classically, the Coulomb branch is a cylinder [3] $R \times S^1$ described by $N-1$ moduli:

$$Y_i \sim \exp\left( \mathbf{\Phi} \cdot \alpha_i 2\pi R/g^2 \right) \, , \tag{B.1}$$

where $\boldsymbol{\Phi}$ is the chiral superfield whose lowest component contains the adjoint scalar $\phi$, $\alpha_i$ are the simple roots, $R$ is the radius of the circle, and $g$ is the 4D gauge coupling. The 3D gauge coupling is defined by

$$\frac{1}{g_3^2} = \frac{2\pi R}{g^2} \ . \tag{B.2}$$

The number of independent Coulomb branch operators depend on the number of singularities where a matter field becomes massless. For an $SU(5)$ gauge theory, the adjoint scalar $\phi$ has VEV (up to gauge transformation) given by

$$\phi = \mathrm{diag}(v_1, v_2, v_3, v_4, v_5) \ , \qquad v_1 + v_2 + v_3 + v_4 + v_5 = 0 \ , \tag{B.3}$$

along with the fundamental Weyl chamber condition (3.16). The Coulomb branch singularities are $v_{2,3,4} = 0$ for massless fundamentals; and $v_1 + v_{4,5} = 0$, $v_2 + v_{3,4,5} = 0$ and $v_3 + v_4 = 0$ for massless antisymmetric matter representations. Higgs branches pinch off the Coulomb branches at those singularities.

In the fundamental Weyl chamber of $SU(5)$ with the matter content shown in Table 1 we summarize the various regions in Table 8. Note that there are zero modes from the antisymmetric

| Region | Zero Modes | Coulomb Operators |
|---|---|---|
| $v_1 > 0 > v_2 > v_3 > v_4 > v_5$ | $\left(F + \overline{F} + 3N_A\right) n_1$ | $\dot{Y}_1, \ \dot{Y}_2, \ \dot{Y}_3, \ \dot{Y}_4$ |
| $v_1 > |v_3| > v_2 > 0 > v_3 > v_4 > v_5$ | $(F + \overline{F})n_2 + 3N_A n_1$ | $\widetilde{Y}_1, \ \widetilde{Y}_2, \ \dot{Y}_3, \ \dot{Y}_4$ |
| $v_1 > |v_4| > v_2 > |v_3| > 0 > v_3 > v_4 > v_5$ | $(F + \overline{F})n_2 + N_A(2n_1 + n_3)$ | $Y_1', \ \widetilde{Y}_2, \ \widetilde{Y}_3, \ \dot{Y}_4$ |
| $v_1 > |v_5| > v_2 > |v_4| > 0 > v_3 > v_4 > v_5$ | $(F + \overline{F})n_2 + N_A(n_1 + n_2 + n_4)$ | $Y_1''', Y_2', \ Y_3', \ \widetilde{Y}_4$ |
| $|v_5| > v_1 > v_2 > |v_4| > 0 > v_3 > v_4 > v_5$ | $(F + \overline{F})n_2 + N_A(n_2 + 2n_4)$ | $Y_1'', Y_2', \ Y_3', \ Y_4''$ |
| $v_1 > v_2 > |v_5| > 0 > v_3 > v_4 > v_5$ | $(F + \overline{F} + 2N_A)n_2 + N_A n_{KK}$ | $\hat{Y}_1, \ \hat{Y}_2, \ Y_3', \ \hat{Y}_4$ |
| $|v_4| > v_1 > v_2 > v_3 > 0 > v_4 > v_5$ | $(F + \overline{F} + 2N_A)n_3 + N_A n_{KK}$ | $\ddot{Y}_1, \ Y_2, \ \ddot{Y}_3, \ \ddot{Y}_4$ |
| $v_1 > |v_5| > |v_4| > v_2 > v_3 > 0 > v_4 > v_5$ | $(F + \overline{F})n_3 + N_A(2n_1 + n_3)$ | $Y_1', \ Y_2, \ Y_3, \ \dot{Y}_4$ |
| $|v_5| > v_1 > |v_4| > v_2 > v_3 > 0 > v_4 > v_5$ | $(F + \overline{F})n_3 + N_A(n_1 + n_3 + n_4)$ | $Y_1, \ Y_2, \ Y_3, \ Y_4$ |
| $|v_5| > v_1 > v_2 > |v_4| > v_3 > 0 > v_4 > v_5$ | $(F + \overline{F})n_3 + N_A(n_2 + 2n_4)$ | $Y_1'', Y_2'', Y_3'', Y_4''$ |
| $|v_5| > v_1 > v_2 > v_3 > |v_4| > 0 > v_4 > v_5$ | $(F + \overline{F})n_3 + 3N_A n_4$ | $Y_1'', Y_2''', Y_3'', Y_4'$ |
| $v_1 > v_2 > v_3 > v_4 > 0 > v_5$ | $\left(F + \overline{F} + 3N_A\right) n_4$ | $Y_1'', Y_2''', Y_3''', Y_4'''$ |

Table 8: Regions of the fundamental Weyl Chamber of $SU(5)$ where $\overline{F} = F + 1$ and $N_A = 1$.

tensor on the KK monopole in some regions. The total number of zero modes in each region is consistent across the regions as required by the fact that the total number of zero modes in $R^3 \times S^1$ for all $N$ monopole solutions including the twisted KK monopole solution should match the number of zero modes of the one 4D instanton given by the Atiyah-Singer index theorem [44].

The Coulomb branch singularities $v_{2,3,4} = 0$, $v_1 + v_{4,5} = 0$, $v_2 + v_{3,4,5} = 0$ and $v_3 + v_4 = 0$ set the boundaries between regions in Table 8. (Not all of the singularities are on the boundary of a given region.) Whenever zero modes jump from monopole $i$ to monopole $j$ when we cross the boundary

from a certain region to another, new independent Coulomb operators for the monopole $i$ and $j$ have to be introduced for the region we are moving into. Continuity is maintained by the fact that near the boundary the monopoles involved in the jumping must be bound together. In the last column in Table 8 we find 30 Coulomb operators in total. Among the 30 Coulomb operators only 12 operators (which do not have a matter zero mode) are lifted. However, the remaining 18 unlifted operators are not all independent, and actually there are only two globally defined operators parametrizing the unlifted Coulomb moduli throughout the Coulomb branch. Integrating out all lifted fields only these globally defined fields and the fields in the Higgs branch appear in the effective superpotential. The first globally defined modulus is $Y \equiv Y_1 Y_2 Y_3 Y_4$ described by the adjoint under the $SU(2)$ corresponding to $\alpha_1 + \alpha_2 + \alpha_3 + \alpha_4$:

$$Y \leftrightarrow \begin{pmatrix} v & & & & \\ & 0 & & & \\ & & 0 & & \\ & & & 0 & \\ & & & & -v \end{pmatrix} . \tag{B.4}$$

Ten regions in Table 8 (all but the two regions that have zero modes on the KK monopole[1]) can reach the $SU(2)$ adjoint (B.4) on their boundary, specifically by taking $v_{2,3,4} \to 0$. That is to say, $Y$ should be continuous across the 10 regions, which imposes 9 constraints. Note that the twisted monopole solution [23] associated with the lowest root (3.22) can be described by the moduli $Y$. The superpotential contribution by the KK monopole is given by

$$W_{KK} = \exp\left(-\frac{4\pi}{g_3^2 R} - \frac{4\pi^2(v_N - v_1)}{g_3^2}\right) = \eta Y , \tag{B.5}$$

where

$$\eta = \exp\left(-\frac{4\pi}{g_3^2 R}\right) = \exp\left(-\frac{8\pi^2}{g_4^2(1/R)}\right) = (R\,\mu_0)^b \exp\left(-\frac{8\pi^2}{g_4^2(\mu_0)}\right) \equiv (R\Lambda)^b . \tag{B.6}$$

There is another globally defined modulus $\widetilde{Y} \equiv \sqrt{(Y_1 Y_2 Y_3 Y_4) Y_2 Y_3}$ [2] corresponding to the adjoint:

$$\widetilde{Y} \leftrightarrow \begin{pmatrix} v & & & & \\ & v & & & \\ & & 0 & & \\ & & & -v & \\ & & & & -v \end{pmatrix} . \tag{B.7}$$

The $\widetilde{Y}$ direction breaks the gauge symmetry from $SU(5) \to (SU(2)^2 \times U(1)^2)/\mathbb{Z}_2$. Only eight regions in Table 8 (all but the first two and the last two regions) can reach to the adjoint (B.7) on its boundary, specifically by taking $v_3 \to 0$, $v_2 + v_4 \to 0$ and $v_1 + v_5 \to 0$ together with $v_2 \to v_1$. Then $\widetilde{Y}$ should be continuous across the eight regions, which imposes other 7 constraints. Thus total 16 constraints reduce the 18 unlifted local Coulomb moduli to two degrees of freedom, and we can describe the unlifted local Coulomb moduli in terms of the globally defined moduli $Y$ and $\widetilde{Y}$ throughout the Coulomb branch.

---

[1]Unlike $SU(N)$ with $N \leq 4$ and an antisymmetric tensor, for $N \geq 5$ the modulus $Y$ is not globally defined in all regions of the Weyl Chamber due to the presence of zero modes on the KK monopole in some regions.

[2]Because of the square root in the definition there can be half-integer charged monopoles [45].

# C  Dynamics in other Weyl chamber regions

Throughout this section we will study multi-monopole configuration in a different region of the fundamental Weyl chamber:

$$|v_5| > v_1 > v_2 > |v_4| > v_3 > 0 > v_4 > v_5 , \tag{C.1}$$

unless otherwise specified. In this region of the fundamental Weyl chamber the first fundamental monopole has two gaugino zero modes, the second monopole has two gaugino zero modes and a zero mode from the antisymmetric tensor doublet $(A_{2,4}, A_{3,4})$ under the $SU(2)$ subgroup corresponding to $\alpha_2$, the third monopole has two gaugino zero modes, $F$ fundamental zero modes, $F+1$ antifundamental zero modes, and the fourth monopole has two gaugino zero modes and two antisymmetric zero modes, one from each of the doublets $(A_{4,1}, A_{5,1})$ and $(A_{4,2}, A_{5,2})$ under the $SU(2)$ subgroup corresponding to $\alpha_4$.

Note that the Coulomb operators, $Y_1''$, $Y_2''$, $Y_3''$, and $Y_4''$, for the region (C.1) are not identical to the Coulomb operators for the region (4.9) described in the main text. For brevity, we will substitute $Y_i'' \to Y_i$ throughout this section.

## C.1  SU(5) with $F = 2$: $\boxvert + 2\square + 3\bar{\square}$

### C.1.1  $F = 2$, $M \gg B_2 \gg \overline{B}_1$

Let's consider the case with hierarchical VEVs:

$$M \gg B_2 \gg \overline{B}_1 . \tag{C.2}$$

For this case we parametrize the antisymmetric VEV as

$$A = \begin{pmatrix} 0 & 0 & 0 & 0 & a \\ 0 & 0 & 0 & 0 & 0 \\ 0 & 0 & 0 & b & 0 \\ 0 & 0 & -b & 0 & 0 \\ -a & 0 & 0 & 0 & 0 \end{pmatrix} , \tag{C.3}$$

and the squark VEVs as

$$Q_{f\alpha} = \begin{pmatrix} 0 & 0 \\ q_{1,2} & 0 \\ 0 & q_{2,3} \\ 0 & 0 \\ 0 & 0 \end{pmatrix} , \quad \overline{Q}_{f,\alpha}^* = \begin{pmatrix} \overline{q}_{1,1}^* & 0 & 0 \\ 0 & 0 & 0 \\ 0 & \overline{q}_{2,3}^* & 0 \\ 0 & 0 & 0 \\ 0 & 0 & \overline{q}_{3,5}^* \end{pmatrix} , \tag{C.4}$$

where $D$-flatness requires

$$2|b|^2 = 2|b|^2 + |q_{2,3}|^2 - |\overline{q}_{2,3}|^2 = |q_{1,2}|^2 = 2|a|^2 - |\overline{q}_{1,1}|^2 = 2|a|^2 - |\overline{q}_{3,5}|^2 . \tag{C.5}$$

For large matter VEVs we can map the composites (see Table 1) onto the classical flat directions: $M \sim \overline{q}_{2,3}\, q_{2,3}$, $B_2 \sim A_{1,5}A_{3,4}q_{1,2}$, $\overline{B}_1 \sim A_{1,5}\overline{q}_{1,1}\overline{q}_{3,5}$.

First we turn on the VEVs $q_{2,3}$ and $\overline{q}_{2,3}$. To be able to do so, we have to restrict the adjoint VEVs to satisfy $v_3 \to 0$, i.e.

$$\phi = \mathrm{diag}(v_1, v_2, 0, v_4, -v_1 - v_2 - v_4) . \tag{C.6}$$

Note that contrary to the Weyl chamber region (4.9), to restrict $v_3 \to 0$ does not require $v_1 + v_5 \to 0$ nor $v_2 + v_4 \to 0$ in region (C.1) and the adjoint VEV is in the Cartan of $SU(4)$, since the matter VEVS break the he gauge symmetry from $SU(5)$ to $SU(4)$. Two matter zero modes on monopole 3 are lifted along with two gaugino zero modes by the Yukawa coupling,

$$g \, q^{*\,3} \lambda_3^3 Q_3 + h.c. \,, \tag{C.7}$$

and the low-energy theory has an antisymmetric, two fundamentals, and two antifundamentals. One fundamental zero mode comes from the components of antisymmetric tensor. The scales are related by

$$\Lambda^{11} = \Lambda_{(4)}^9 \, q_{2,3} \, \bar{q}_{2,3} \,. \tag{C.8}$$

The unbroken Cartan elements are:

$$Q_1 \;=\; \frac{1}{2} \, \mathrm{diag}(1, -1, 0, 0, 0) \tag{C.9}$$

$$Q_{2+3} \;=\; \frac{1}{2} \, \mathrm{diag}(0, 1, 0, -1, 0) \tag{C.10}$$

$$Q_4 \;=\; \frac{1}{2} \, \mathrm{diag}(0, 0, 0, 1, -1) \,. \tag{C.11}$$

The broken Cartan elements is:

$$X = Q_1 + 2Q_2 - 2Q_3 - Q_4 \;=\; \frac{1}{2} \, \mathrm{diag}(1, 1, -4, 1, 1) \,. \tag{C.12}$$

There is a confined composite monopole comprised of monopole 2 and monopole 3 to be neutral under the broken generator. The 2+3 composite monopole has two gaugino zero modes, two fundamental zero modes, and two antifundamental zero modes unlifted, so it cannot contribute to the superpotential. The fundamental monopole 1 has two zero modes unlifted, so it does contribute to the superpotential. Monopole 4 has four zero modes, so it doesn't contribute to the superpotential. We have a superpotential:

$$W = \eta_4 Y_{SU(4)} + \frac{1}{Y_1} = \eta_4 Y_{2+3} Y_1 Y_4 + \frac{1}{Y_1} = \eta Y + \frac{1}{Y_1} \,, \tag{C.13}$$

where $Y_{2+3} = Y_1 Y_2 M$. A sketch of the 2+3 composite monopole is shown in Fig. 12. In this diagram we explicitly show "resonance" diagrams where the gauginos lifted by the Yukawa coupling (C.7) are explicitly shown without simplifying the diagram by moving fermion zero modes through the string/flux tube.

We can arrive this same $SU(4)$ effective theory and adjoint VEVs (C.6) from another fundamental Weyl chamber region:

$$|v_5| \geq v_1 \geq v_2 \geq |v_4| \geq 0 \geq v_3 \geq v_4 \geq v_5 \,. \tag{C.14}$$

In this region (anti-)fundamental zero modes live on monopole 2 and zero modes from the antisymmetric tensor remain on the monopole 2 and monopole 4 (with multiplicity 2). With the same argument as above, turning on VEVs $q_{2,3}$ and $\bar{q}_{2,3}$ with $v_3 \to 0^-$ gives rise to the superpotential (C.13) and the 2+3 composite with two gaugino zero modes, two fundamental zero modes, and two antifundamental zero modes similar to Fig. 12.

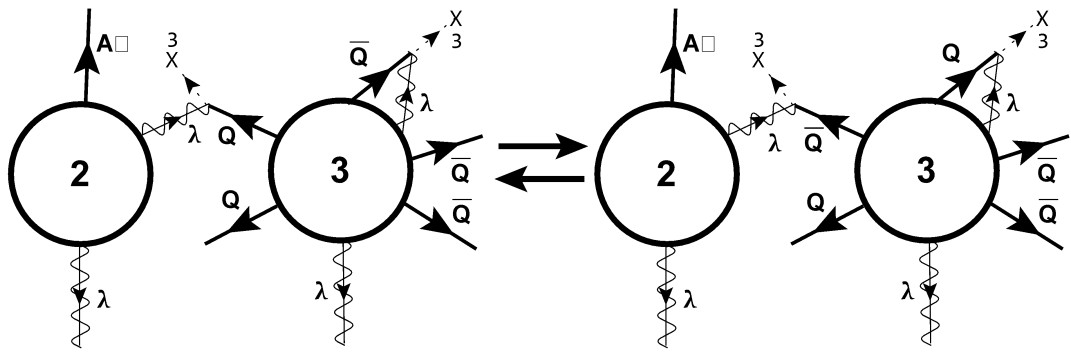

Figure 12: A sketch of multi-monopole composite with $\boxminus + 2\square + 3\bar{\square}$ when $M$ is large. The gauge group breaks from $SU(5)$ to $SU(4)$. Two "resonance" diagrams equivalently form the 2+3 composite. The anti-symmetric and squark VEVs, (C.3) and (C.4), are represented as X with their indices. $A_\square$ represents the components of anti-symmetric tensor which transform as a fundamental representation under the unbroken $SU(4)$ subgroup.

Turning $a$, $b$ and $q_{1,2}$ as well with $v_{2,4} \to 0$ and $v_1 + v_5 \to 0$ then breaks the gauge symmetry from $SU(4)$ to $SU(2)$ leaving two antifundamentals with scales related by

$$\Lambda_{(4)}^9 = \Lambda_{(2)}^5 \, a^2 \, b \, q_{1,2} \ . \tag{C.15}$$

The adjoint VEV is in the Cartan of $SU(2)_a$, as shown in (D.10).

The unbroken Cartan element is:

$$Q_{1+2+3+4} \quad = \quad \frac{1}{2} \operatorname{diag}(1, 0, 0, 0, -1) \ . \tag{C.16}$$

The additional broken Cartan elements are:

$$Q_{2+3} \quad = \quad \frac{1}{2} \operatorname{diag}(0, 1, 0, -1, 0) \tag{C.17}$$

$$Q_{1-4} \quad = \quad \frac{1}{2} \operatorname{diag}(1, -1, 0, -1, 1) \ . \tag{C.18}$$

The fundamental monopoles are all confined to form a neutral composite under the broken generators, and monopole 1 and 4 join the 2+3 composite turning into a 1+2+3+4 composite monopole. The 1+2+3+4 composite monopole has two gaugino zero modes and two antifundamental zero modes unlifted, so it cannot contribute to the superpotential. The corresponding superpotential is

$$W = \eta_2 Y_{SU(2)} = \eta_4 Y_1 Y_2 Y_3 Y_4 \, q_{2,3} \, \bar{q}_{2,3} = \eta Y \ , \tag{C.19}$$

where $Y_{SU(2)} = Y_{2+3} Y_1 Y_4 \, a \, b^2 \, q_{1,2}$. A sketch of the 1+2+3+4 monopole is shown in Fig. 13.

Finally turning on $\bar{q}_{1,1}$ and $\bar{q}_{3,5}$ VEVs breaks $SU(2)$ completely and the KK monopole joins with the 1+2+3+4 composite to form an instanton with two unlifted gaugino zero modes. A sketch of the KK+1+2+3+4 instanton is shown in Fig. 14. The superpotential

$$W = \frac{\eta Y}{Y_2 Y_3 \, q_{2,3} \, \bar{q}_{2,3} \, Y_1 Y_4 \, a^2 \, b \, q_{1,2} \, \bar{q}_{1,1} \, \bar{q}_{3,5}} = \frac{\eta}{B_2 \bar{B}_1 M} \ . \tag{C.20}$$

which matches (5.10).

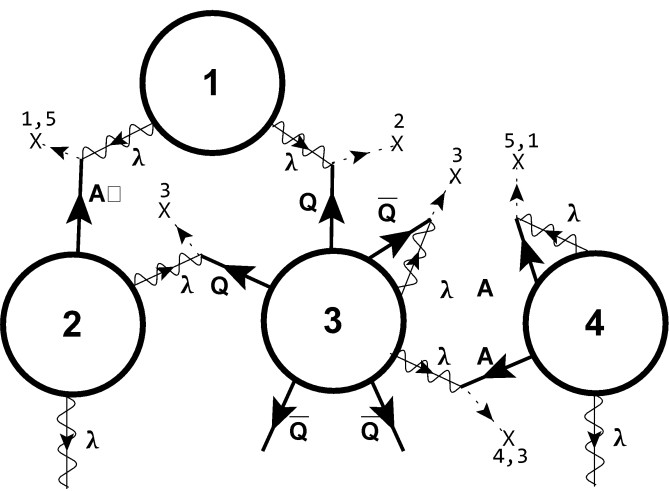

Figure 13: A sketch of multi-monopole composite with $\boxminus + 2\square + 3\overline{\square}$ when $M \gg B_2$ are large. The gauge group breaks from $SU(5)$ to $SU(2)$. Only one of the "resonance" diagrams is shown. The anti-symmetric and squark VEVs, (C.3) and (C.4), are represented as X with their indices. $A_\square$ represents the components of anti-symmetric tensor which transform as a fundamental representation under $SU(4)$.

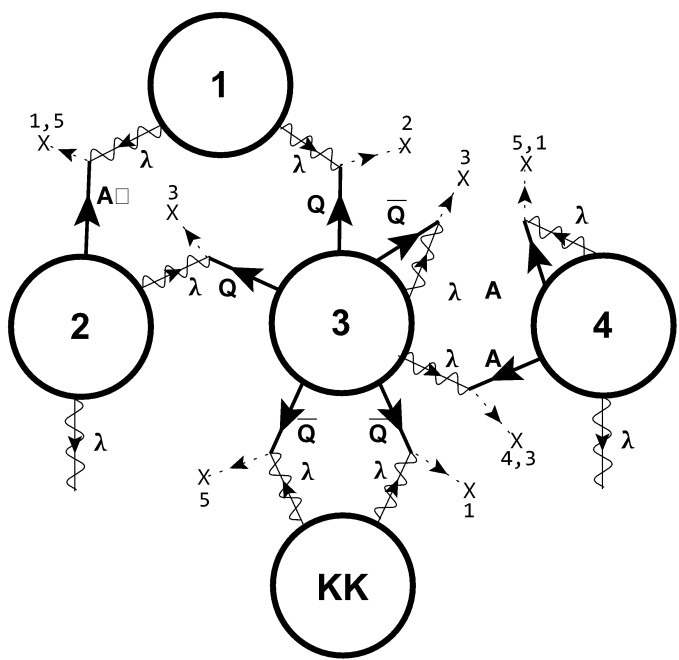

Figure 14: A sketch of multi-monopole composite with $\boxminus + 2\square + 3\overline{\square}$ when $M \gg B_2 \gg \overline{B}_1$ are large. The gauge group is completely broken. Only one of the "resonance" diagrams is shown. The anti-symmetric and squark VEVs, (C.3) and (C.4), are represented as X with their indices.

## C.2  SU(5) with $\boxminus + \square + 2\overline{\square}$

We will consider the case with hierarchical VEVs,

$$\overline{B}_1 \gg B_2 \gg \Lambda,\, 1/R \,. \tag{C.21}$$

For this case we will investigate the dynamics with the antisymmetric VEV parametrizing as

$$
A = \begin{pmatrix}
0 & 0 & 0 & 0 & a \\
0 & 0 & 0 & 0 & 0 \\
0 & 0 & 0 & b & 0 \\
0 & 0 & -b & 0 & 0 \\
-a & 0 & 0 & 0 & 0
\end{pmatrix} , \tag{C.22}
$$

and the squark VEVs as

$$
Q_\alpha = \begin{pmatrix} 0 \\ q_2 \\ 0 \\ 0 \\ 0 \end{pmatrix} , \quad \overline{Q}^*_{f,\alpha} = \begin{pmatrix} 0 & 0 \\ 0 & 0 \\ \overline{q}^*_{1,3} & 0 \\ 0 & \overline{q}^*_{2,4} \\ 0 & 0 \end{pmatrix} , \tag{C.23}
$$

where $D$-flatness requires

$$
2|a|^2 = |q_2|^2 = 2|b|^2 - |\overline{q}_{1,3}|^2 = 2|b|^2 - |\overline{q}_{2,4}|^2 . \tag{C.24}
$$

For large matter VEVs we can map the composites (see Table 1) onto the classical flat directions: $\overline{B}_1 \sim A_{3,4}\overline{q}_{1,3}\,\overline{q}_{2,4}$, $B_2 \sim A_{1,5}A_{3,4}q_2$.

A large $\overline{B}_1$ turned on with $v_{3,4} \to 0$ breaks the gauge symmetry from $SU(5)$ to $SU(3)$ leaving a fundamental and an antifundamental. The scales are related by

$$
\Lambda^{12} = \Lambda^8_{(3)}\, b^2\overline{q}_{1,3}\,\overline{q}_{2,4} . \tag{C.25}
$$

The unbroken Cartan elements are::

$$
Q_1 = \frac{1}{2}\operatorname{diag}(1, -1, 0, 0, 0) \tag{C.26}
$$

$$
Q_{2+3+4} = \frac{1}{2}\operatorname{diag}(0, 1, 0, 0, -1) . \tag{C.27}
$$

The broken $U(1)$ generators are:

$$
Q_3 = \frac{1}{2}\operatorname{diag}(0, 0, 1, -1, 0)
$$

$$
X = 2Q_1 + 4Q_2 + Q_3 - 2Q_4 = \frac{1}{2}\operatorname{diag}(2, 2, -3, -3, 2) . \tag{C.28}
$$

There is a confined composite monopole comprised of monopoles 2, 3, and 4 which is neutral under the broken generators (C.28). The 2+3+4 composite has two gaugino zero modes, one fundamental zero mode and one antifundamental zero mode unlifted, so it cannot contribute to the superpotential. The fundamental monopole 1 has two gaugino zero modes, so it contributes to the superpotential. A sketch of the 2+3+4 composite monopole is shown in Fig. 15.

The corresponding superpotential is

$$
W = \eta_3 Y_1 Y_{2+3+4} + \frac{1}{Y_1} = \eta Y + \frac{1}{Y_1} , \tag{C.29}
$$

where $Y_{2+3+4} = Y_2 Y_3 Y_4\, b^2 \overline{q}_{1,3}\,\overline{q}_{2,4}$.

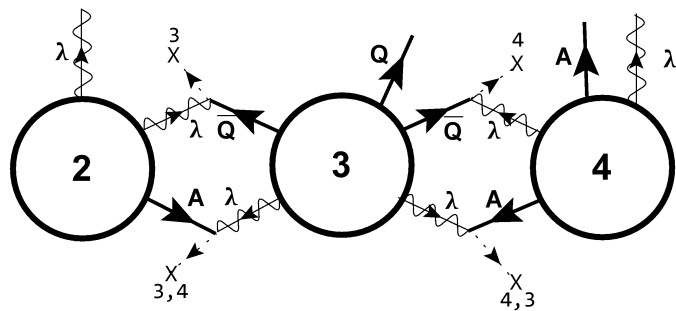

Figure 15: A sketch of the multi-monopole composite with $\boxminus + \square + 2\bar{\square}$ when $\overline{B}_1$ is large. The gauge group breaks to $SU(3)$. The antisymmetric and squark VEVs, (C.22) and (C.23), are represented as a cross with their color indices.

Turning on the $a$ and $q_2$ VEVs further breaks the gauge symmetry from $SU(3)$ to $SU(2)$ leaving no matter fields. The scales are related by

$$\Lambda_3^8 = \Lambda_{(2)}^6 \, a \, q_2 \; . \tag{C.30}$$

The unbroken Cartan element is:

$$Q_{1+2+3+4} \;\; = \;\; \frac{1}{2} \, \mathrm{diag}(1, 0, 0, 0, -1) \; . \tag{C.31}$$

The additional broken $U(1)$ generator is:

$$Q_{1-2-3-4} \;\; = \;\; \frac{1}{2} \, \mathrm{diag}(1, -2, 0, 0, 1) \; . \tag{C.32}$$

The fundamental monopole 1 joins with the 2+3+4 composite to form a confined 1+2+3+4 composite monopole. A sketch of the 1+2+3+4 monopole constructed by turning on $\overline{B}_1$ and $B_2$ VEVs in sequence are shown in Fig. 16. The 1+2+3+4 monopole has two unlifted zero modes, so it does contributes to the superpotential. The superpotential is

$$W \;\; = \;\; \eta_2 Y_{SU(2)} + \frac{1}{Y_{SU(2)}} = \eta_3 Y_1 Y_{2+3+4} + \frac{1}{Y_1 Y_{2+3+4} \, a \, q_2} \tag{C.33}$$

$$= \;\; \eta Y + \frac{1}{Y \, a \, q_2 \, b^2 \, \overline{q}_{1,3} \overline{q}_{2,4}} \; , \tag{C.34}$$

where $Y_{SU(2)} = Y_1 Y_{2+3+4} \, a \, q_2$. Integrating out the Coulomb branch moduli we recover (5.33), so the calculations in the two different regions agree.

# D    Other Hierarchical Patterns for $F = 2$

### D.0.1    $F = 2$, $\overline{B}_1 \gg B_2 \gg M$

Let's next consider the case with hierarchical VEVs,

$$\overline{B}_1 \gg B_2 \gg M \; , \tag{D.1}$$

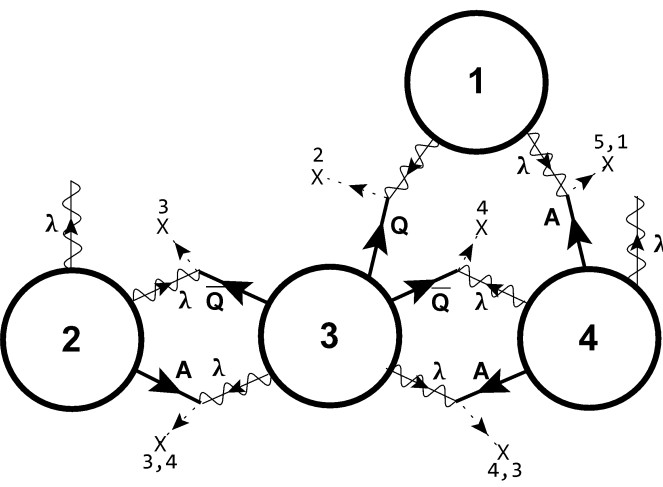

Figure 16: A sketch of the multi-monopole composite with $\boxvert + \square + 2\overline{\square}$ when $\overline{B}_1 \gg B_2$ are large. The gauge group breaks to $SU(2)$. The antisymmetric and squark VEVs, (C.22) and (C.23), are represented as a cross with their color indices.

with the antisymmetric and squark VEVs parametrized as in (5.6) and (5.7). The $D$-flatness condition is

$$2|a|^2 = 2|a|^2 + |q_{1,1}|^2 - |\bar{q}_{1,1}|^2 = |q_{2,3}|^2 = 2|b|^2 - |\bar{q}_{2,2}|^2 = 2|b|^2 - |\bar{q}_{3,4}|^2 \, , \tag{D.2}$$

and for large matter VEVs we can map the composites (see Table 1) onto the classical flat directions: $\overline{B}_1 \sim A_{2,4}\bar{q}_{2,2}\bar{q}_{3,4}$, $B_2 \sim A_{1,5}A_{2,4}q_{2,3}$, $M \sim \bar{q}_{1,1}q_{1,1}$. The VEV $b$ is invariant under $SU(3)_a \times SU(2)_b$ with unbroken Cartan generators

$$Q_{1+2} = \frac{1}{2} \operatorname{diag}(1, 0, -1, 0, 0) \, , \tag{D.3}$$

$$Q_{3+4} = \frac{1}{2} \operatorname{diag}(0, 0, 1, 0, -1) \, , \tag{D.4}$$

$$Q_{2+3} = \frac{1}{2} \operatorname{diag}(0, 1, 0, -1, 0) \, . \tag{D.5}$$

The broken $U(1)$ generator is

$$X = 2(Q_1 - Q_4) - Q_2 + Q_3 = \frac{1}{2} \operatorname{diag}(2, -3, 2, -3, 2) \, . \tag{D.6}$$

The $\bar{q}_{2,2}$ and $\bar{q}_{3,4}$ VEVs then leaves only an unbroken $SU(3)_a$ gauge invariance. The unbroken Cartan elements are:

$$Q_{1+2} = \frac{1}{2} \operatorname{diag}(1, 0, -1, 0, 0) \, , \tag{D.7}$$

$$Q_{3+4} = \frac{1}{2} \operatorname{diag}(0, 0, 1, 0, -1) \, . \tag{D.8}$$

The embedding of representations is shown in Table 9. The two triplet representations of the antisymmetric and two anti-triplets from the antifundamentals are eaten by the broken gauge supermultiplets, so the low-energy $SU(3)_a$ theory has only two fundamentals and two antifundamentals with one of the antifundamentals being a descendant of the original antisymmetric. In

$$
\begin{array}{lllll}
SU(5) & \rightarrow & SU(3)_a \times SU(2)_b & \rightarrow & SU(3)_a \\
5 & \rightarrow & (3,1) + (1,2) & \rightarrow & 3 + 1 + 1 \\
10 & \rightarrow & (3,2) + (\bar{3},1) + (1,1) & \rightarrow & 3 + 3 + \bar{3} + 1 \\
24 & \rightarrow & (8,1) + (1,3) + (3,2) + (\bar{3},2) + (1,1) & \rightarrow & 8 + 3 + 3 + \bar{3} + \bar{3} + 4 \cdot 1
\end{array}
$$

Table 9: Embedding of representations $SU(3)_a \times SU(2)_b$ and $SU(3)_a$ into representations of $SU(5)$.

other words the low-energy theory is $SU(3)_a$ with two flavors. The scales of the $SU(5)$ theory and the low-energy $SU(3)_a$ theory are related by

$$
\Lambda^{11} = \Lambda_{(3a)}^7 \, b^2 \bar{q}_{2,2} \, \bar{q}_{3,4} \ . \tag{D.9}
$$

There are two confined $U(1)$ charges corresponding to the broken generators (D.6) and (D.5). The neutral composites are monopole 1 with monopole 2 and monopole 3 with monopole 4. The KK monopole is neutral under the broken generators.

In region (4.9), turning on VEVs for $\overline{B}_1$ (i.e. $b$, $\bar{q}_{2,2}$ and $\bar{q}_{3,4}$) restricts the adjoint VEV to the remainder of the Cartan of $SU(3)_a$: $v_{2,3,4} \to 0$ and $v_1 + v_5 \to 0$, i.e.

$$
\phi = \mathrm{diag}(v_1, 0, 0, 0, -v_1) \ . \tag{D.10}
$$

Turning to the VEVs for $B_2$ (i.e. $a$ and $q_{2,3}$) we need to remember that in the effective $SU(3)_a$ theory the VEV $a$ is in the anti-color corresponding to the color of $q_{2,3}$. Thus the VEVs $a$ and $q_{2,3}$ break $SU(3)_a$ to $SU(2)_a$ leaving one fundamental and one antifundamental. There is no further restriction on the adjoint VEV, (D.10), since it was already forced to be in the Cartan of $SU(2)_a$.

The scale of the $SU(2)_a$ effective theory is given by

$$
\Lambda_{(3a)}^7 = \Lambda_{(2a)}^5 \, a \, q_{2,3} \ . \tag{D.11}
$$

The unbroken Cartan element is:

$$
Q_{1+2+3+4} = \frac{1}{2} \mathrm{diag}(1, 0, 0, 0, -1) \ , \tag{D.12}
$$

while the new broken $U(1)$ generator is:

$$
Q_{1+2-3-4} = \frac{1}{2} \mathrm{diag}(1, 0, -2, 0, 1) \ . \tag{D.13}
$$

To be neutral under all three broken $U(1)$ generators, the monopoles 1, 2 3, and 4 are confined together. The KK monopole is neutral under the broken generators. The 1+2+3+4 composite monopole is shown in Fig. 6; it has four unlifted zero modes, so it cannot contribute to the superpotential. The superpotential is thus:

$$
W = \eta_{(2a)} Y^{(2a)} = \eta Y \ , \tag{D.14}
$$

where $Y^{(2a)} = Y_1 Y_2 Y_3 Y_4 \, a b^2 \, q_{2,3} \bar{q}_{2,2} \, \bar{q}_{3,4} = Y_1 Y_2 Y_3 Y_4 \, B_2 \overline{B}_1$.

The $q_{1,1}$ and $\bar{q}_{1,1}$ VEVs break $SU(2)_a$ completely and the KK monopoles joins with the 1+2+3+4 composite, forming an instanton with two unlifted zero modes. A sketch of the KK+1+2+3+4 instanton is shown in Fig. 7. The instanton superpotential is

$$
W_{F=2} = \frac{\eta Y}{Y_1 Y_2 Y_3 Y_4 b^2 \bar{q}_{2,2} \, \bar{q}_{3,4} \, a \, q_{2,3} \, q_{1,1} \, \bar{q}_{1,1}} = \frac{\eta}{B_2 \overline{B}_1 M} \ , \tag{D.15}
$$

which matches (5.10).

**D.0.2** $F = 2$, $\overline{B}_1 \gg M \gg B_2$

The case with hierarchical VEVs

$$\overline{B}_1 \gg M \gg B_2 \tag{D.16}$$

is similar to the previous case. For this case we use the antisymmetric VEV

$$A = \begin{pmatrix} 0 & 0 & 0 & a & 0 \\ 0 & 0 & b & 0 & 0 \\ 0 & -b & 0 & 0 & 0 \\ -a & 0 & 0 & 0 & 0 \\ 0 & 0 & 0 & 0 & 0 \end{pmatrix}, \tag{D.17}$$

which is invariant under $SU(2)_c \times SU(2)_d$, and the squark VEVs

$$Q_{f\alpha} = \begin{pmatrix} 0 & 0 \\ 0 & 0 \\ 0 & 0 \\ q_{1,4} & 0 \\ 0 & q_{2,5} \end{pmatrix}, \quad \overline{Q}^*_{f,\alpha} = \begin{pmatrix} 0 & 0 & 0 \\ \overline{q}^*_{1,2} & 0 & 0 \\ 0 & \overline{q}^*_{2,3} & 0 \\ 0 & 0 & \overline{q}^*_{3,4} \\ 0 & 0 & 0 \end{pmatrix}, \tag{D.18}$$

where $D$-flatness requires

$$2|a|^2 = 2|b|^2 - |\overline{q}_{1,2}|^2 = 2|b|^2 - |\overline{q}_{2,3}|^2 = 2|a|^2 + |q_{1,4}|^2 - |\overline{q}_{3,4}|^2 = |q_{2,5}|^2 . \tag{D.19}$$

For large matter VEVs we can map the composites (see Table 1) onto the classical flat directions: $\overline{B}_1 \sim A_{2,3}\,\overline{q}_{1,2}\overline{q}_{2,3}$, $M \sim \overline{q}_{3,4}\,q_{1,4}$, $B_2 \sim A_{1,4}A_{2,3}\,q_{2,5}$.

Again in region (4.9) of the fundamental Weyl chamber turning on VEVs for $\overline{B}_1$ requires the restriction $v_{2,3,4} \to 0$ and $v_1 + v_5 \to 0$. The large $\overline{B}_1$ VEV (corresponding to $b, q_{1,2}$, and $\overline{q}_{2,3}$ VEVs) breaks the gauge symmetry to an $SU(3)_c$ subgroup with Cartan generators:

$$Q_{1+2+3} = \frac{1}{2}\,\text{diag}(1, 0, 0, -1, 0) , \tag{D.20}$$

$$Q_4 = \frac{1}{2}\,\text{diag}(0, 0, 0, 1, -1) , \tag{D.21}$$

and there is a composite monopole made of 1+2+3. Turning on a VEV for $M$ further breaks the gauge symmetry to $SU(2)_a$ and there is a composite monopole made of 1+2+3+4. Turning on $B_2$ forces a composite of 1+2+3+4 and the KK monopole resulting in the usual instanton superpotential (5.21).

Note that by turning on the VEVs for $\overline{B}_1$ the gauge symmetry breaks from $SU(5)$ to $SU(3)$, however, the adjoint VEV starting from the fundamental Weyl chamber region (4.9) is forced to be in the Cartan of $SU(2)$. In Appendix C we provide a discussion for composite monopoles in another Weyl chamber region of Table 8 which allows for the adjoint VEV to be in the Cartan of $SU(3)_c$ on its boundary.

**D.0.3** $F = 2$, $M \gg \overline{B}_1, B_2$

Let's consider the case with hierarchical VEVs,

$$M \gg \overline{B}_1 \gg B_2 . \tag{D.22}$$

For this case we use the antisymmetric VEV parameterized as

$$A = \begin{pmatrix} 0 & 0 & 0 & 0 & 0 \\ 0 & 0 & 0 & b & 0 \\ 0 & 0 & 0 & 0 & a \\ 0 & -b & 0 & 0 & 0 \\ 0 & 0 & -a & 0 & 0 \end{pmatrix} , \tag{D.23}$$

and the squark VEVs

$$Q_{f\alpha} = \begin{pmatrix} q_{1,1} & 0 \\ 0 & 0 \\ 0 & q_{2,3} \\ 0 & 0 \\ 0 & 0 \end{pmatrix} , \quad \overline{Q}^*_{f,\alpha} = \begin{pmatrix} 0 & 0 & 0 \\ \overline{q}^*_{1,2} & 0 & 0 \\ 0 & \overline{q}^*_{2,3} & 0 \\ 0 & 0 & \overline{q}^*_{3,4} \\ 0 & 0 & 0 \end{pmatrix} , \tag{D.24}$$

where $D$-flatness requires

$$2|a|^2 = 2|a|^2 + |q_{2,3}|^2 - |\overline{q}_{2,3}|^2 = |q_{1,1}|^2 = 2|b|^2 - |\overline{q}_{1,2}|^2 = 2|b|^2 - |\overline{q}_{3,4}|^2 . \tag{D.25}$$

For large matter VEVs we can map the composites (see Table 1) onto the classical flat directions: $M \sim \overline{q}_{2,3}\, q_{2,3}$, $\overline{B}_1 \sim A_{2,4}\overline{q}_{1,2}\overline{q}_{3,4}$, $B_2 \sim A_{3,5}A_{2,4}q_{1,1}$.

First turning on the $M$ VEV (i.e. $q_{2,3}$ and $\overline{q}_{2,3}$), the gauge symmetry breaks from $SU(5)$ to $SU(4)$, and the low-energy theory has an antisymmetric, two fundamentals and two antifundamentals (see Table 4 for the embedding of representations). Note that one of the fundamentals of the low energy theory arises from the components of antisymmetric tensor, the VEV of this fundamental corresponds to $a$ in our parameterization (D.23). The scales are related by

$$\Lambda^{11} = \Lambda^9_{(4)}\, q_{2,3}\, \overline{q}_{2,3} . \tag{D.26}$$

The unbroken Cartan elements are:

$$Q_1 = \frac{1}{2}\, \mathrm{diag}(1, -1, 0, 0, 0) \tag{D.27}$$

$$Q_{2+3} = \frac{1}{2}\, \mathrm{diag}(0, 1, 0, -1, 0) \tag{D.28}$$

$$Q_4 = \frac{1}{2}\, \mathrm{diag}(0, 0, 0, 1, -1) , \tag{D.29}$$

while the broken generator is:

$$X = Q_1 + 2Q_2 - 2Q_3 - Q_4 = \frac{1}{2}\, \mathrm{diag}(1, 1, -4, 1, 1) . \tag{D.30}$$

Monopoles 2 and 3 are confined so as to be neutral under the broken generator. In region (4.9) of the fundamental Weyl chamber, turning on VEVs for $q_{2,3}$ and $\overline{q}_{2,3}$ further restricts the adjoint

VEV to be in the Cartan of $SU(4)$, so $v_3 \to 0$, $v_2 + v_4 \to 0$ and $v_1 + v_5 \to 0$. Thus the adjoint VEV has the form

$$\phi = \text{diag}(v_1, v_2, 0, -v_2, -v_1) \ . \tag{D.31}$$

Next turning on a VEV for $\overline{B}_1$ (i.e $b$, $\bar{q}_{1,2}$, and $\bar{q}_{3,4}$), we see that the antisymmetric VEV $b$ is invariant under an $Sp(4)$ subgroup, while the $\bar{q}$ VEVs reduce this to an unbroken $SU(2)_a$ gauge symmetry leaving just two doublets in the effective gauge theory. The scales of the $SU(4)$ theory and the low-energy $SU(2)$ theory are related by

$$\Lambda_{(4)}^9 = \Lambda_{(2a)}^5 \, b^2 \, \bar{q}_{1,2} \, \bar{q}_{3,4} \ . \tag{D.32}$$

A composite monopole $1+2+3+4$ is confined so as to be neutral under the broken generators, while the KK monopole is neutral under the broken generators. The $1+2+3+4$ monopole has four unlifted zero modes (see Fig. 6), so it cannot contribute to the superpotential. The superpotential is:

$$W = \eta_{(2a)} Y^{(2a)} = \eta Y \ , \tag{D.33}$$

where $Y^{(2a)} = Y_1 Y_2 Y_3 Y_4 \, \bar{q}_{2,3} \, q_{2,3} \, b^2 \, \bar{q}_{1,2} \, \bar{q}_{3,4} = Y_1 Y_2 Y_3 Y_4 \, M \overline{B}_1$. The associated extended Dynkin diagrams for this breaking pattern are shown in Fig. 17.

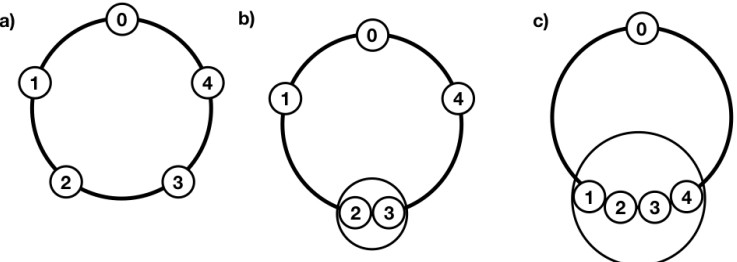

Figure 17: The extended Dynkin diagrams for breaking patterns of $SU(5)$: a) $SU(5)$, b) $SU(4)$, and c) $SU(2)_a$.

Finally turning on the $a$ and $q_{1,1}$ VEVs (i.e. tuning on the VEVs for the remaining two doublets) breaks $SU(2)$ completely and confines the KK monopole with the $1+2+3+4$ composite to form and instanton with two unlifted gaugino zero modes. and the final superpotential is

$$W_{F=2} = \frac{\eta Y}{Y_1 Y_2 Y_3 Y_4 \, q_{2,3} \, \bar{q}_{2,3} \, b \, \bar{q}_{1,2} \, \bar{q}_{3,4} \, a \, b \, q_{1,1}} = \frac{\eta}{B_2 \overline{B}_1 M} \ . \tag{D.34}$$

which again matches (5.10).

The case with hierarchical VEVs

$$M \gg B_2 \gg \overline{B}_1 \tag{D.35}$$

is similar. For this case we use the antisymmetric VEV

$$A = \begin{pmatrix} 0 & 0 & 0 & 0 & a \\ 0 & 0 & 0 & 0 & 0 \\ 0 & 0 & 0 & b & 0 \\ 0 & 0 & -b & 0 & 0 \\ -a & 0 & 0 & 0 & 0 \end{pmatrix} \ , \tag{D.36}$$

and squark VEVs

$$Q_{f\alpha} = \begin{pmatrix} 0 & 0 \\ q_{1,2} & 0 \\ 0 & q_{2,3} \\ 0 & 0 \\ 0 & 0 \end{pmatrix} , \quad \overline{Q}^*_{f,\alpha} = \begin{pmatrix} \overline{q}^*_{1,1} & 0 & 0 \\ 0 & 0 & 0 \\ 0 & \overline{q}^*_{2,3} & 0 \\ 0 & 0 & 0 \\ 0 & 0 & \overline{q}^*_{3,5} \end{pmatrix} , \tag{D.37}$$

where $D$-flatness requires

$$2|b|^2 = 2|b|^2 + |q_{2,3}|^2 - |\overline{q}_{2,3}|^2 = |q_{1,2}|^2 = 2|a|^2 - |\overline{q}_{1,1}|^2 = 2|a|^2 - |\overline{q}_{3,5}|^2 . \tag{D.38}$$

For large matter VEVs we can map the composites (see Table 1) onto the classical flat directions: $M \sim \overline{q}_{2,3}\, q_{2,3}$, $B_2 \sim A_{1,5} A_{3,4} q_{1,2}$, $\overline{B}_1 \sim A_{1,5} \overline{q}_{1,1} \overline{q}_{3,5}$.

Turning on VEVs for $M$ breaks $SU(5)$ to $SU(4)$. As before there is confined multi-monopole 2+3, and further gauge symmetry breaking from turning on VEVs for $B_2$ breaks the gauge symmetry to $SU(2)_a$, producing a 1+2+3+4 composite multi-monopole. Turning on the $B_1$ VEV breaks the gauge symmetry completely, and an instanton is formed, and we again arrive at the superpotential (5.21).

Note that by turning on the VEVs for $M$ the gauge symmetry breaks from $SU(5)$ to $SU(4)$, however, the adjoint VEV starting from the fundamental Weyl chamber region (4.9) is forced to be in the Cartan of $Sp(4)$. In Appendix. C we provide a discussion for composite monopoles in another Weyl chamber region (see Table 8) which allows an adjoint VEV to be in the Cartan of $SU(4)$ on its boundary.

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
