# Peer review of "Chiral Gauge Theories on $R^3 \times S^1$ and SUSY Breaking"

_SciPost Physics_

## Round 1 · Referee Report · Anonymous · 2019-9-30

Strengths

This is a nice paper presenting a detailed analysis of the dynamics of a chiral SU(5) theory compactified on the circle. It lends further support to both known ADS superpotentials in 4d and to pre-ADS superpotential in the circle-compactified theory. Further, there is new evidence in favor of the supersymmetry-breaking scenario of the one-flavor SU(5) theory.

Weaknesses

The only weakness I can think of is the very technical nature of the subject. I am not sure it can be remedied, it certainly requires familiarity/reading of previous papers (incl. ones by the authors). I do not hold this remark against the paper.

Report

I recommend this paper for publication. The main reason is the new essentially semiclassical argument for supersymmetry breaking in the F=0 theory presented in the paper.

---

## Round 1 · Referee Report · Anonymous · 2019-10-23

Strengths

1 - Nice analysis of a class of SUSY gauge theories compactified on a circle.

2 - Nice discussion of monopole effects in these theories.

Weaknesses

No real weaknesses, except that more motivation for this analysis would be nice to see in the introduction.

Report

This paper presents a thorough analysis of monopole effects and their relation to SUSY breaking in SU(5) ${\cal N}=1$ SYM theories with antisymmetric tensor, fundamental and antifundamental matter chiral multiplets. By compactifying one dimension on a circle, the authors are able to interpolate between results in 3D and 4D. Some of the other usual techniques of interpolating between theories are employed, for example by integrating out flavors. Where independent analyses are available, the authors compare their results with the existing literature.

One nice feature of the manuscript is the appendices, where details of some of the hard labor appears, including analysis of zero modes on monopole configurations and exploration of the different regions in the moduli space of these theories.

In my opinion this is a fine analysis of a class of SUSY gauge theories, and while the phenomenological motivation for such analyses is becoming less obvious with time, this work is a worthy addition to the literature.

Requested changes

I really have only one minor suggestion and two questions that the authors might consider responding to by adding a sentence or two to the text.

1- Unless I missed it, the notation for Coulomb branch operators in the effective superpotentials of Table 2 are not explained until much later in the text and appendices. It would be helpful either to provide brief definitions of notation in either the caption to Table 2 or the surrounding text, or else to point to the relevant places elsewhere in the text where the notation is subsequently defined.

2 - For my own edification, as part of the discussion of the interpolation between the compactified and uncompactified theories, I would have appreciated some additional comments regarding which, if any, topological effects might persist in the decompactified limit.

3 - Even if the objective importance of dynamical SUSY breaking and monopole confinement is self-evident, a gesture to possible phenomenological applications of these models or techniques would be nice to see.

---

## Editorial Decision

editor-in-charge_assigned